# Membrane-cytoskeletal crosstalk mediated by myosin-I regulates adhesion turnover during phagocytosis

Sarah R. Barger [1], Nicholas S. Reilly[2], Maria S. Shutova[3], Qingsen Li[4], Paolo Maiuri [4], John M. Heddleston [5], Mark S. Mooseker[6], Richard A. Flavell [7,8], Tatyana Svitkina[3], Patrick W. Oakes [2,9], Mira Krendel [1] & Nils C. Gauthier[4]

Phagocytosis of invading pathogens or cellular debris requires a dramatic change in cell shape driven by actin polymerization. For antibody-covered targets, phagocytosis is thought to proceed through the sequential engagement of Fc-receptors on the phagocyte with antibodies on the target surface, leading to the extension and closure of the phagocytic cup around the target. We find that two actin-dependent molecular motors, class 1 myosins myosin 1e and myosin 1f, are specifically localized to Fc-receptor adhesions and required for efficient phagocytosis of antibody-opsonized targets. Using primary macrophages lacking both myosin 1e and myosin 1f, we find that without the actin-membrane linkage mediated by these myosins, the organization of individual adhesions is compromised, leading to excessive actin polymerization, slower adhesion turnover, and deficient phagocytic internalization. This work identifies a role for class 1 myosins in coordinated adhesion turnover during phagocytosis and supports a mechanism involving membrane-cytoskeletal crosstalk for phagocytic cup closure.

[1] Cell and Developmental Biology Department, State University of New York Upstate Medical University, Syracuse 13210 NY, USA. [2] Department of Physics, University of Rochester, Rochester 14627 NY, USA. [3] Department of Biology, University of Pennsylvania, Philadelphia 19104 PA, USA. [4] IFOM, FIRC Institute of Molecular Oncology, Milan 20139, Italy. [5] Advanced Imaging Center, Howard Hughes Medical Institute Janelia Research Campus, Ashburn 20147 VA, USA. [6] Molecular, Cellular and Developmental Biology, Yale University, New Haven 06520 CT, USA. [7] Department of Immunobiology, Yale University School of Medicine, New Haven 06519 CT, USA. [8] Howard Hughes Medical Institute, Yale University, New Haven 06519 CT, USA. [9] Department of Biology, University of Rochester, Rochester 14627 NY, USA. Correspondence and requests for materials should be addressed to M.K. (email: krendelm@upstate.edu) or to N.C.G. (email: nils.gauthier@ifom.eu)

Phagocytosis is a critical immune response that requires coordinated adhesion, membrane rearrangement, and dynamic remodeling of the actin cytoskeleton[1]. Internalization via Fcγ receptors (FcRs), which bind the conserved domain of immunoglobulins, involves several stages, beginning with the clustering of FcRs that activate downstream signaling pathways to induce assembly of an actin-rich, cup-like structure (the phagocytic cup) that surrounds the target[2]. The plasma membrane of the phagocytic cup is extended by the force of branched actin polymerization and, if a target is particularly large, additional membrane from intracellular stores is added to the cup by exocytosis[3]. Cup fusion results in a de novo membrane-bound organelle (the phagosome), which is shuttled further into the cell for processing and degradation[4]. While the signaling pathways that link FcR clustering to the initiation of F-actin assembly are well understood[5], extension and closure of the phagocytic cup, which requires regulated actin polymerization and coactive membrane deformation, remains enigmatic.

Past studies have revealed that phagocytosis is both driven and regulated by mechanical forces[6]. For a successful phagocytic event, the force of actin polymerization within the extending arms of the phagocytic cup must overcome mechanical properties of the cell itself, namely membrane and cortical tension. However, as a phagocyte ingests a target, both membrane and cortical tension increase[7–9], and these properties in turn can regulate addition of new membrane through exocytosis. Over the course of phagocytosis, macrophages experience a steep increase in membrane tension, which triggers exocytosis of intracellular membrane stores that increase cell surface area for internalization[9]. However, it is unknown how or if this change in membrane tension affects the actin assembly required for phagocytic cup closure.

The longstanding model of phagocytic cup closure involves F-actin assembly at discrete FcR adhesions between the phagocyte and the IgG-coated particle, with subsequent cup extension driven by the formation of additional Fc receptor-IgG bonds in a zipper-like fashion along the target[10]. Here, we report that two class 1 myosins, myosin 1e (myo1e) and myosin 1f (myo1f), small monomeric actin-based motors that can bind to the actin cytoskeleton through their motor domains and the plasma membrane through their tails, are associated with Fc-receptor adhesions and control membrane tension and organization at these sites throughout phagocytosis. Using a myo1e/f double knockout (dKO) mouse model, we find that macrophages lacking these myosins assemble phagocytic cups of clumped and disorganized actin, exhibit slower FcR adhesion turnover and, as a result, are deficient at internalizing targets. By tethering membrane around FcR adhesion sites, myo1e/f work to spatially confine actin assembled via FcR signaling. Overall, this work describes a biophysical component precisely controlling actin dynamics to promote extension and closure of the phagocytic cup.

## Results

### Myo1e/f localize at the phagocytic cup and drive cup closure.
To examine the localization of myo1e and myo1f throughout phagocytosis, we used fluorescence microscopy on both live and fixed cells. RAW264.7 macrophages transfected with fluorescently tagged myo1e or myo1f and actin-labeling constructs were challenged to engulf 6 μm latex beads opsonized in mouse IgG. We found that during bead ingestion, myo1e was recruited to the cup and colocalized with the extending belt of phagocytic actin, as previously observed[11], yet slightly preceded actin at the leading edge of the cup (arrowheads) (Fig. 1a–d, Supplementary Movie 1). Myo1f exhibited similar behavior during engulfment (Supplementary Fig. 1a-d, Supplementary Movie 2) and when co-expressed, the two myosins showed near perfect colocalization at

the cup tip (Fig. 1e). These observations were reinforced by immunostaining of endogenous myo1e and actin during phagocytosis in primary murine bone marrow-derived macrophages (BMDM) (Fig. 1f). These results indicate a clear recruitment and enrichment of myo1e/f at the progressing cup during phagocytosis. Although colocalizing with the actin belt, both myo1e/f are uniquely situated at the leading edge of the cup, particularly evident near the time of cup closure (Fig. 1d and Supplementary Fig. 1d, arrowheads, Supplementary Movies 1 and 2).

After establishing a clear colocalization between actin and myo1e/f during phagocytosis, we next investigated the relationship between these myosins and plasma membrane phospholipids. During phagocytic internalization, lipid composition within the cup undergoes a series of changes, which parallel and likely regulate sequential stages in actin assembly and dynamics[12]. PtdIns(4,5)$P_2$ (PIP2) populates the extending arms of the phagocytic cup and is replaced by PtdIns(3,4,5)$P_3$ (PIP3) as the cup closes[13,14]. As myo1e and myo1f contain putative Pleckstrin Homology (PH) domains, capable of interacting with both PIP2 and PIP3[15,16], we tested whether the two myosins occupied the same regions as these lipids. RAW macrophages were co-transfected with fluorescent myo1e/f and the EGFP-tagged lipid sensors PLCδ-PH, for PtdIns(4,5)$P_2$, and AKT-PH, for PtdIns(3,4,5)$P_3$ (Supplementary Fig. 2a)[13,17]. As AKT-PH is known to bind both PtdIns(3,4,5)$P_3$ and PtdIns(3,4)$P_2$, we also utilized TAPP1 as an exclusive PI(3,4)$P_2$ sensor[18] (Supplementary Fig. 2b). For a negative control, we used PKCδ-C1, a sensor for diacylglycerol[19], known to be enriched in sealed phagosomes[20]. By assessing the percentage of cells that displayed myosin and lipid sensor co-localization, we concluded that myo1e and myo1f co-localized predominantly with PtdIns(3,4,5)$P_3$ at the phagocytic cup (Supplementary Fig. 2b). Co-localization with PtdIns(4,5)$P_2$ and PI(3,4)$P_2$ was also observed, but at lower frequencies. To test the dependence of myosin localization at the cup on phospholipid accumulation, we assessed cup recruitment of myo1e and myo1f following treatment with LY294002, an inhibitor of PI3K, thus blocking PtdIns(3,4,5)$P_3$ production at the phagocytic cup[21,22]. Both myo1e and myo1f still localized to the phagocytic cup, although to a lesser extent (Supplementary Fig. 2c-d). Together, these results show that myo1e/f may interact with both cytoskeleton and membrane at their location within the cup through actin and phosphoinositide binding, respectively, but do not solely depend on PtdIns(3,4,5)$P_3$ for their recruitment.

Based on the unique localization of the two myosins at the phagocytic cup, we set out to test whether myo1e and myo1f were required for efficient phagocytosis. We isolated BMDM from wild-type mice, myo1e KO mice[23], and myo1f KO mice[24]. Both KO strains were created using a constitutive (germline) knockout approach, so that cultured macrophages completely lack myo1e or myo1f (Fig. 1g). Given the similarity in myo1e/f domain structure and the likelihood of functional redundancy, as suggested by our localization results, we also generated a dKO mouse to best understand the role of these proteins in phagocytosis. To measure phagocytic activity, we challenged BMDMs to engulf 6 μm IgG-coated latex beads. We identified un-internalized beads by fixing cells and staining beads prior to cell permeabilization (Supplementary Fig. 3a). By quantifying the percentage of cells that engulfed at least one latex bead, we found that macrophages lacking either myo1e or myo1f performed similar to control cells (Fig. 1h). However, when both myosins were knocked out, macrophages appeared to be engulfing beads more slowly, with a significantly lower percentage of phagocytosing cells at each time point (Fig. 1h–i). This result suggests myo1e and myo1f are required for efficient phagocytosis, yet perform functionally redundant roles, and thus all subsequent experiments with BMDMs were conducted using WT and dKO macrophages.

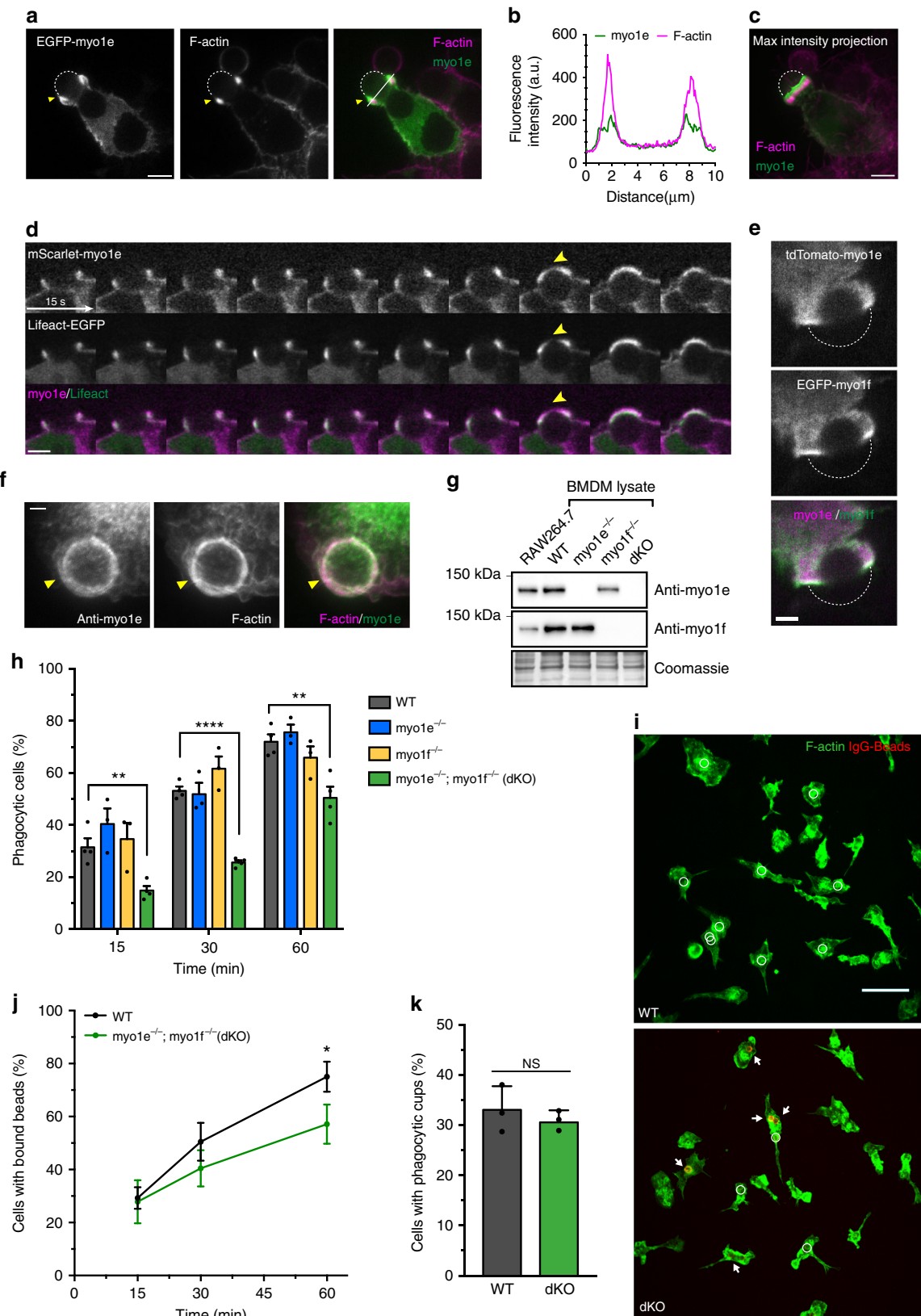

**Early steps in phagocytosis do not require myo1e/f.** With myo1e and myo1f binding both actin filaments and membrane lipids, the lack of these myosins could potentially affect FcR clustering activity at the plasma membrane or the ability of cells to assemble actin to form phagocytic cups. To test whether these mechanisms could account for the observed defect

in phagocytosis, we first assessed downstream signaling from FcRs by antibody crosslinking[25]. BMDM were exposed to soluble anti-FcR antibody at 4 °C, followed by the addition of a secondary antibody to cross-link the first. The cells were then incubated at 37 °C for different periods of time to initiate signaling. We then probed for phosphorylated Syk kinase, an early FcR-mediated

**Fig. 1** Myo1e/f are required for efficient phagocytosis. **a** Confocal section of EGFP-myo1e-transfected RAW264.7 macrophage engulfing 6 μm IgG-coated bead and stained with fluorescent phalloidin. Yellow arrowhead indicates the phagocytic cup, dotted line outlines the bead. Scale bar, 5 μm. **b** Line scan of EGFP-myo1e and F-actin intensity along the line in **a**. **c** Maximum intensity projection of **a** shows that myo1e precedes actin at the cup leading edge. Scale bar, 5 μm. **d** Time-lapse montage of RAW macrophage expressing mScarlet-myo1e and Lifeact-EGFP engulfing 8 μm IgG-coated bead. Yellow arrowhead points to myo1e preceding F-actin, particularly at cup closure. Scale bar, 5 μm. **e** Myo1e and myo1f colocalize at the edge of phagocytic cup. Confocal section of tdTomato-myo1e/EGFP-myo1f-transfected RAW macrophage engulfing 6 μm IgG-coated bead (dotted line). Scale bar, 2 μm. **f** WT BMDM staining with anti-myo1e and phalloidin shows that endogenous myo1e colocalizes with F-actin at the phagocytic cup (arrow) formed around 6 μm IgG-coated bead. Cup is open, facing upward. Total intensity projection of a confocal z-stack. Scale bar, 2 μm. **g** Western blots of myo1e/f in RAW264.7 cells and WT, myo1e$^{-/-}$, myo1f$^{-/-}$, and dKO BMDM. Equal protein loading verified by Coomassie Blue staining. **h** Percentage (mean ± SEM) of cells that internalized at least one 6 μm IgG-coated bead at 15, 30, and 60 min. Data from three to four experiments. Analysis of 15–30 FOV resulted on average in 1200 cells per genotype quantified per experiment (15 min: $p = 0.005$; 30 min: $p < 0.0001$; 60 min: $p = 0.006$, unpaired $t$-tests). **i** Images of phagocytosis assay at 15 min. BMDM are stained with phalloidin (green) and un-internalized beads (arrows) are stained red. White circles denote internalized beads (not visible in the fluorescence image). Scale bar, 50 μm. **j** Percentage of cells (mean ± SD) that bound at least one bead during phagocytosis time course experiments described in **h** (15 min: $p = 0.7625$; 30 min: $p = 0.09$; 60 min: $p = 0.029$, unpaired $t$-tests). **k** Percentage of cells (mean ± SD) that formed actin-based phagocytic cups at 10 min ($p = 0.44$, unpaired $t$-test). Data from three independent experiments. Analysis of 10–18 FOV per experiment resulted in 1026 WT and 1557 dKO cells quantified

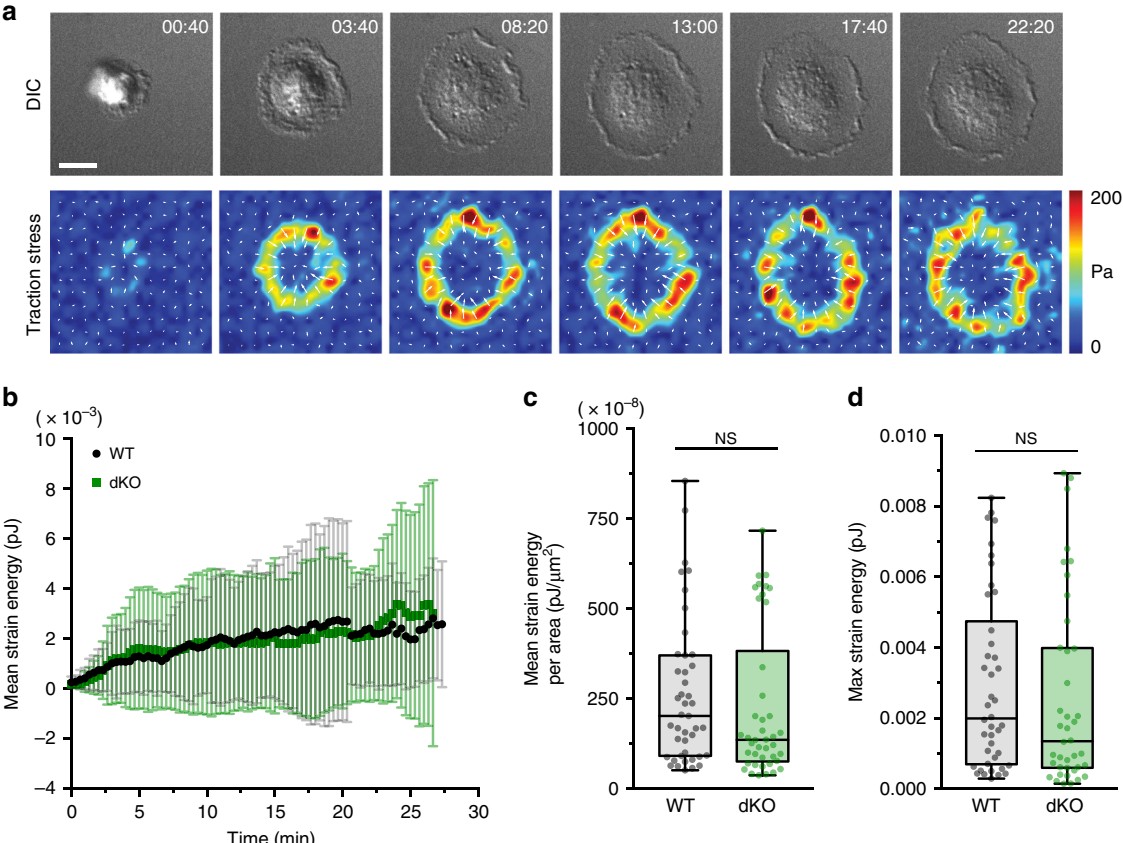

**Fig. 2** Myo1e and myo1f do not contribute to phagocytic contractility. **a** Representative time-lapse montage of BMDM (dKO macrophage) performing frustrated phagocytosis and exhibiting traction forces. Differential interference contrast (DIC) imaging (top) above traction force map (bottom). The magnitude of the brightness in the traction map corresponds to the magnitude of the stress (i.e., a pixel value of 50 = 50 Pa), with the pixel intensity values color-coded as indicated by the color wedge on the right. Scale bar, 10 μm. See also Supplementary Movie 3. **b** Graph of mean strain energy (±SD) over time during spreading. WT and dKO macrophages performed frustrated phagocytosis for TFM measurements. Data from two independent experiments ($n = 44$ WT and 42 dKO cells). **c** Graph of mean strain energy per unit of cell area. Box and whisker plot shows median, 25th and 75th percentile, with error bars depicting maximum and minimum data points. ($n = 43$ WT and 42 dKO cells). Two outliers have been removed ($p = 0.43$, unpaired $t$-test). **d** Graph of maximum strain energy measured over 30 min of cell spreading. Box and whisker plot shows median, 25th and 75th percentile, with error bars depicting maximum and minimum data points. ($n = 42$ WT and 40 dKO cells). Four outliers have been removed ($p = 0.48$, unpaired $t$-test)

signaling player, and phosphorylation of Akt and ERK, known to occur later. Levels of phosphorylated Syk, Akt, and downstream ERK signaling in dKO cells were indistinguishable from those in WT macrophages (Supplementary Fig. 3b-c). We next tested whether FcR surface presentation or interactions with the

antibody-coated targets were affected. Flow cytometry showed that FcR surface expression was unaffected in dKO cells (Supplementary Fig. 3d). In addition, the percentage of cells that were simply associated with a bead, whether external (bound) or internal (ingested), was similar between WT and dKO BMDM

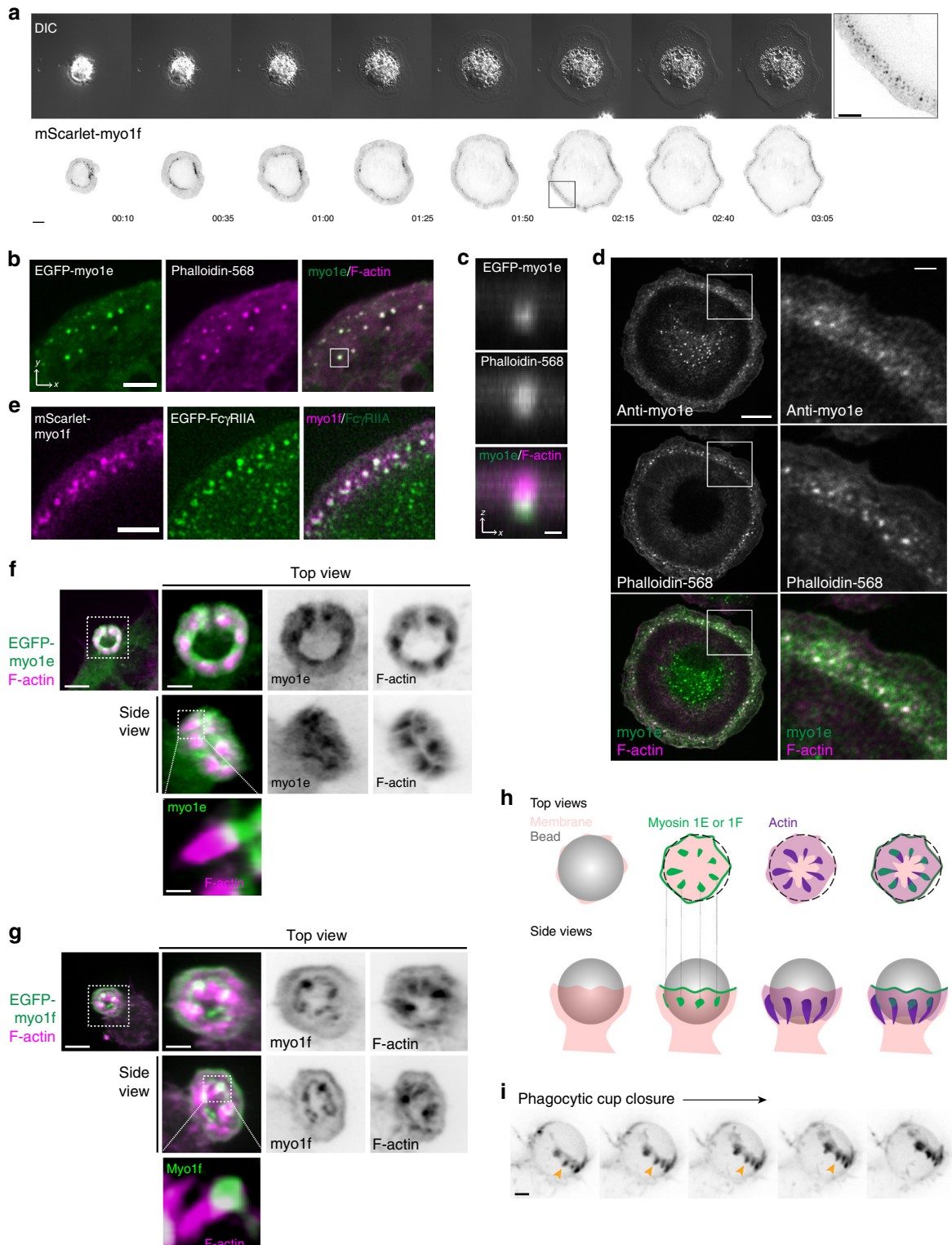

(Fig. 1j). Notably, at 15 min, when a 50% defect in bead ingestion was observed (Fig. 1h), we detected no statistically significant difference in bead association between the WT and dKO macrophages (Fig. 1j). Both the unchanged receptor density and efficiency of early bead binding suggested that myosin-I loss did not affect FcR behavior on the cell surface.

We next questioned whether the lack of myo1e/f was affecting phagocytic cup formation. WT and dKO BMDM were exposed to IgG-coated latex beads for 10 min, then fixed and stained with fluorescently labeled phalloidin to identify phagocytic cups. Surprisingly, the percentage of cells with phagocytic cups did not differ between WT and dKO cells (Fig. 1k). Given the successful bead binding and formation of phagocytic cups in dKO macrophages, the observed lag in complete bead internalization can only be due to a defect during cup progression and/or closure in the absence of myo1e/f.

**Fig. 3** Myo1e and myo1f localize to FcR-actin adhesions during frustrated phagocytosis. **a** Time-lapse montage of mScarlet-myo1f-transfected RAW macrophage (inverted) conducting frustrated phagocytosis and imaged by DIC (top) and TIRF microscopy (bottom). Scale bar, 10 μm. Zoom of boxed region depicting myo1f puncta is shown on the right. Scale bar, 5 μm. **b** F-actin colocalizes with myo1e at F-actin puncta. Confocal section of spreading edge of RAW macrophage expressing EGFP-myo1e and stained with Alexa Fluor-568-phalloidin. Scale bar, 5 μm. See also Supplementary Fig. 5a. **c** xz projection of the boxed region in **b** showing myo1e at the base of the phagocytic adhesion. Scale bar, 0.5 μm. **d** Primary macrophages form myo1e-enriched actin punctae during frustrated phagocytosis. WT BMDM spreading on IgG-coated coverslips for 10 min were fixed and stained with myo1e antibody and fluorescently labeled phalloidin. Scale bar, 10 μm. Zoom of boxed region shown on the right. Scale bar, 2 μm. **e** Myo1f puncta colocalize with Fc receptors. TIRF image of the spreading edge of RAW macrophage co-expressing mScarlet-myo1f and EGFP-FcγRIIA. Scale bar, 5 μm. **f, g** Phagocytic cups contain plumes of F-actin emanating from discrete myosin-I adhesion sites. 3D representations of phagocytic cup of RAW macrophages transfected with either EGFP-myo1e (**f**) or EGFP-myo1f (**g**) and counter-stained with fluorescently labeled phalloidin (bead not outlined). Scale bar, 5 μm; inset scale bar, 2 μm; zoom scale bar, 250 nm. **h** Graphical representation of myo1e/f and F-actin localization at the phagocytic cup (top view and side view). For the top view, gray bead has been replaced by dotted outline to allow visualization inside the cup. **i** Actin adhesions within the phagocytic cup move over target during phagocytic internalization. Maximum intensity projection time-lapse montage of RAW macrophage transfected with mEmerald-Lifeact (inverted) engulfing a 7 μm IgG-coated latex bead, imaged by lattice light sheet microscopy. Orange arrowheads point to the actin plumes dynamically progressing along the bead behind the edge of the phagocytic cup. Scale bar, 4 μm

**Myo1e/f do not affect contraction in frustrated phagocytosis.** As myo1e has been previously proposed to drive contraction of the phagocytic cup[26], and contractility is known to be required for the completion of phagocytosis[27], the inability of dKO cells to efficiently close their phagocytic cups may be the result of reduced contractility applied by the macrophage on the target. We set out to test whether macrophages lacking myo1e/f are deficient in contractile force production using traction force microscopy (TFM), which has recently been validated to measure contractility during a 2D frustrated phagocytosis assay[28]. During frustrated phagocytosis, cells spread on an IgG-coated substrate, which stimulates actin and membrane rearrangement that faithfully mimics the 3D process[29–32]. For TFM, flat but deformable polyacrylamide gels were prepared for frustrated phagocytosis by first coating with BSA followed by anti-BSA mouse antibody. WT and dKO macrophages were then dropped on to the gels and imaged for 30 min while spreading (Fig. 2a, Supplementary Movie 3). Over time, the mean strain energy produced by the dKO cells did not significantly differ from that of WT macrophages (Fig. 2b, c). Quantification of the maximum strain energy also did not reveal any significant differences in contractile activity that could be attributed to myo1e/f (Fig. 2d). Thus, myo1e/f appear not to contribute to the contractile force applied by macrophages to the target during FcR-mediated phagocytosis, as measured by this 2D assay.

**Myo1e and myo1f do not participate in focal exocytosis.** We next hypothesized that lack of focal exocytosis, the local insertion of intracellular membrane at the phagocytic cup, may account for the decreased phagocytic efficiency in the absence of myosin-Is. Focal exocytosis allows cells to fully engulf large targets, such as the 6 μm beads used in our study. Because class 1 myosins are known to localize to endosomes and exocytic vesicles[33,34], we hypothesized that myo1e/f may be mediating membrane vesicle delivery at the phagocytic cup explaining the slower cup closure observed in the dKO BMDM. To test this hypothesis, we challenged cells to engulf smaller targets (2 μm IgG-coated latex beads), which should not require focal exocytosis activity[21]. Despite the smaller target size, we found that myo1e still localized to the phagocytic cup (Supplementary Fig. 4a). This agrees with our finding that this motor protein is recruited in a PI3K-independent manner, which is known to be dispensable for the ingestion of smaller targets[35]. We found that dKO macrophages still exhibited slower uptake of 2 μm beads, with no differences in bead binding (Supplementary Fig. 4b-c). To further investigate membrane dynamics, we utilized a lipid-based dye, N-(3-Triethylammoniumpropyl)-4-(4-(dibutylamino) styryl) pyridinium dibromide (FM 1-43), to measure increases in cell surface area,

regardless of the specific membrane source added, during frustrated phagocytosis (Supplementary Movie 4). As we have previously shown using this assay, macrophages undergoing frustrated phagocytosis increase their cell surface area by ~40%, which is reflected as an increase in FM 1-43 intensity[9]. Using WT and dKO macrophages, we observed no difference in FM 1-43 intensity over the course of cell spreading (Supplementary Fig. 4d-f). In past studies, defects in focal exocytosis and subsequent failure to extend the phagocytic cup has been demonstrated by a reduction in spread cell area on IgG or on surfaces coated with bacterial proteins[36–38]. We thus quantified spread cell area during 25 min of frustrated phagocytosis, yet found no difference in dKO cell area compared to WT macrophages (Supplementary Fig. 4g). Moreover, the velocity of the leading edge during cell spreading was also measured and showed no difference between WT and dKO cells (Supplementary Fig. 4h). Finally, we used RAW macrophages to appraise myo1e/f localization with respect to focal exocytosis markers, such as VAMP3[39]. We detected no co-localization between myo1e/f and VAMP3, finding that class 1 myosins appear spatially separated from focal exocytosis machinery within the cup (Supplementary Fig. 4i). Together, these data affirm that myo1e/f are not involved in focal exocytosis.

**Myo1e/f localize to FcR-actin adhesions.** Having ruled out target binding, initial cup formation, and early pseudopod extension, as well as contractile force generation and focal exocytosis, as potential explanations for the phagocytic defect affecting dKO BMDM, we next investigated actin dynamics within the phagocytic cup. As early pseudopod protrusion may not be affected, given that initial spreading velocity is similar in WT and dKO cells (Supplementary Fig. 4h), we were particularly interested in actin turnover following initial cup formation when continued actin polymerization is required to push the membrane forward to complete internalization. To better visualize phagosomal actin, we imaged cells performing frustrated phagocytosis by total internal reflection fluorescence microscopy (TIRFM). While performing frustrated phagocytosis, macrophages are stimulated to form dynamic circular actin waves, cleared of prominent cortical actin, that imitate the actin polymerization in the extending arms of the phagocytic cup[32]. Similar to F-actin waves in other cell types[40], these structures propagate along the ventral plasma membrane, with actin polymerization at their front followed by depolymerization at their rear. In macrophages performing frustrated phagocytosis, these waves are composed of numerous F-actin puncta and undergo spatial oscillations[32]. Using RAW macrophages transfected with fluorescent myo1e/f, we found that both proteins colocalize (Supplementary Fig. 5a)

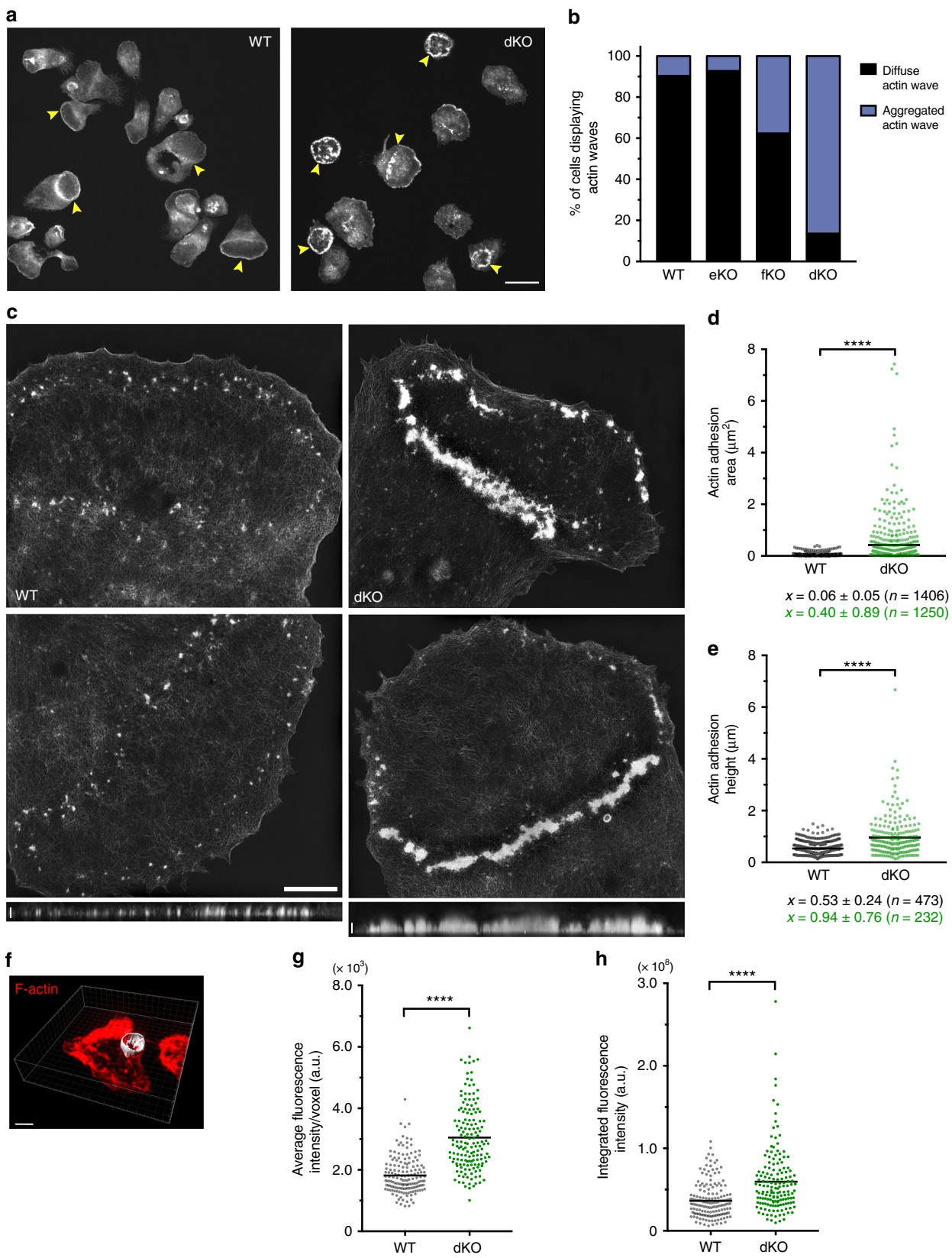

with the punctate actin that makes up the wave (Fig. 3a, b, Supplementary Movie 5). We observed the same localization using primary macrophages, probing for endogenous myo1e (Fig. 3d). While myo1e/f appear to colocalize with actin wave punctae in *xy* confocal sections, *xz* projections reveal myo1e/f to

be concentrated at the base of these structures (Fig. 3b, c). We tested whether these myosin-I/actin puncta were associated with FcR clusters by transfecting cells with EGFP-FcγRIIA. As has been previously observed, macrophages spreading on IgG formed discrete FcR clusters[41,42]. We found that myo1f and FcγRIIA

**Fig. 4** dKO BMDM form disorganized actin adhesions and exhibit denser phagocytic cups. **a** Representative confocal image of WT and dKO macrophages performing frustrated phagocytosis. Cells spread for 10 min, then were fixed and stained with fluorescently labeled phalloidin. Yellow arrows point to circular actin waves. Scale bar, 20 μm. **b** Graph of mean percentage of cells forming diffuse or aggregated actin waves in WT, myo1e$^{-/-}$ (eKO), myo1f$^{-/-}$ (fKO), and myo1e$^{-/-}$; myo1f$^{-/-}$ (dKO) BMDM. Data from 2 to 4 independent experiments. At least 10 FOVs judged blindly per experiment resulting in > 300 cells quantified. **c** Representative structured illumination microscopy (SIM) images of actin wave in BMDM. Scale bar, 5 μm. *xz* projections of the entire cell in the lower images are shown below. Scale bar, 1 μm. **d** Area of individual actin adhesions in WT and dKO macrophages measured using 3D-SIM. Data pooled from two independent experiments. Mean (± SD) annotated below ($n = 1406$ WT and 1250 dKO individual adhesions from at least 20 cells per genotype, ****$p < 0.0001$, unpaired *t*-test). **e** Height of individual actin adhesions in WT and dKO macrophages measured by 3D-SIM. Data pooled from two independent experiments. Mean (± SD) below ($n = 476$ WT and 230 dKO adhesions from 10 cells per genotype, ****$p < 0.0001$, unpaired *t*-test). **f** Representative image of segmented phagocytic cup (gray) for fluorescence intensity measurement of F-actin (red). BMDM were challenged with 6 μm IgG-coated latex beads, fixed and stained with fluorescently labeled phalloidin. Phagocytic cup was measured by 3D reconstruction using Imaris software. Scale bar, 5 μm. **g**, **h** Mean (black line) fluorescence intensity (**g**) and integrated fluorescence intensity (**h**) of F-actin in phagocytic cups of WT and dKO macrophages. Data pooled from three independent experiments ($n = 163$ WT and 156 dKO cups, ****$p < 0.0001$, unpaired *t*-test)

distinctly co-localized at these structures (Fig. 3e, Supplementary Movies 6 and 7), suggesting myo1e/f have a specific role at the membrane at FcR-IgG adhesions.

Since this is the first report of myosin-Is as components of FcR adhesions, and macrophages express multiple myosin-I isoforms, we set out to test whether the localization of myo1e/f at FcR adhesions was unique compared to other class 1 myosins or reflective of a common role for myosin-I in phagocytosis. Myo1c and myo1g have previously been implicated in phagocytosis (myo1g) or detected on macrophage phagosomes (myo1c)[43,44]. They are known as short-tailed myosins as they lack the additional tail domains of myo1e/f (TH2 and SH3 domains, see Supplementary Fig. 5b). In transfected RAW cells, neither myo1c nor myo1g localized to FcR adhesions during frustrated phagocytosis (Supplementary Fig. 5c, d), and their localization at the phagocytic cup differed from that of the long-tailed myosin-Is, myo1e/f (Supplementary Fig. 5c, d). To test whether the additional tail domains of myo1e/f (TH2 and SH3) were responsible for their specific localization to the actin wave, we used a series of truncated myosin constructs and found that myo1e/f localization was dependent on the presence of the TH2 domain in the tail (Supplementary Fig. 6a-d). In addition, a point mutation that disrupts motor domain function also prevented myosin localization[45] (Supplementary Fig. 6b, c). Thus, long-tailed myosins (myo1e/f) appear to have a distinctive role at FcR adhesions.

Although Fcγ receptors are not classically defined as adhesion molecules, the actin wave observed during frustrated phagocytosis or the inside of the phagocytic cup are undoubtedly adhesive structures. IgG coating promotes phagocyte adhesion to surfaces[46,47] and the tight seal created by macrophages performing frustrated phagocytosis was believed to create a closed compartment for cytolytic activity[48]. To test whether the actin wave of macrophages performing frustrated phagocytosis is adhesive, we attempted to detach cells exhibiting actin waves using a micropipette (Supplementary Movie 8). Using RAW macrophages transfected with Lifeact to label F-actin, we found that cells exhibiting actin waves were considerably harder to detach and appeared firmly attached to the substrate by the actin wave structure. By comparison, transfected cells that were not forming an actin wave were easily detached. This observation supports the adhesive nature of the waves, which consist of FcR-myo1e/f-actin punctae. In addition to detecting myo1e/f in punctate adhesions on planar IgG-coated surfaces, we also observed myo1e/f in distinct punctate structures at the bead-membrane interface within 3D phagocytic cups (Fig. 3f–g, Supplementary Movies 9-10). Extending from these myosin-I puncta were plumes of actin polymerization. We hypothesize that these myo1e/f-occupied regions represent distinct FcR adhesion sites within the cup that support assembled F-actin (Fig. 3h). By

live-cell imaging using lattice light sheet microscopy, these adhesions appear to move along the bead during phagocytic internalization (Fig. 3i, Supplementary Movie 11).

**Loss of myo1e/f leads to denser phagocytic cups**. To determine the role of myo1e/f at FcR adhesions, we conducted the frustrated phagocytosis assay using primary macrophages lacking myo1e/f. Similar to the RAW cells, BMDM spread on IgG-coated coverslips and formed circular actin waves cleared of prominent cortical actin. The primary cells rarely formed such structures when allowed to spread on uncoated glass or fibronectin-coated coverslips (Supplementary Fig. 7a-b). Strikingly, we found that actin waves in the dKO macrophages appeared thicker and more clumped than in WT cells (Fig. 4a). The fraction of WT and dKO cells that formed actin waves did not differ (Supplementary Fig. 7c), yet the majority of dKO cells assembled waves of clumped or aggregated actin, which were rarely observed in WT or single KO macrophages (Fig. 4b; see also Supplementary Fig. 7d). To examine the actin waves at higher resolution, we used structured illumination microscopy (SIM), which can improve resolution by a factor of 2 in 3D (to ~100 nm in $x/y$ and 300 nm in $z$). SIM revealed actin waves of WT macrophages to be composed of fairly uniformly sized actin punctae (Fig. 4c), similar to the actin adhesion clusters in the RAW macrophages, and reminiscent of macrophage podosomes[49]. This delicate organization was completely lost in the absence of myo1e/f, as dKO cells formed waves of clumped and aggregated F-actin (Fig. 4c). Using 3D-SIM, we measured these structures and found that actin adhesion clusters in the dKO cells were not only significantly larger in area, but also in height (Fig. 4d, e). To verify that this aggregated actin phenotype was not specific to the frustrated phagocytosis assay, we quantified actin fluorescence in fixed phagocytic cups of WT and dKO macrophages. Similar to the frustrated phagocytosis data, macrophages lacking myo1e/f assembled phagocytic cups with simply more actin (Fig. 4f–h). In an effort to rescue the aggregated actin wave phenotype of the dKO macrophages, we treated cells with small doses of Latrunculin A, which prevents F-actin polymerization by sequestering G-actin. However, such efforts proved unsuccessful, with increasing drug concentration leading only to the inhibition of cell spreading (Supplementary Fig. 8). We also hypothesized that the use of Jasplakinolide might stabilize the actin waves of WT macrophages to phenocopy the actin wave morphology in the dKO cells, yet this was also not observed (Supplementary Fig. 8).

It is generally accepted that the F-actin within the phagocytic cup is primarily nucleated by the Arp2/3 complex, although evidence of formin-assisted nucleation also exists[50,51]. Indeed, staining the actin waves with Arp3 antibody showed discrete puncta in WT macrophages and significant Arp2/3 recruitment

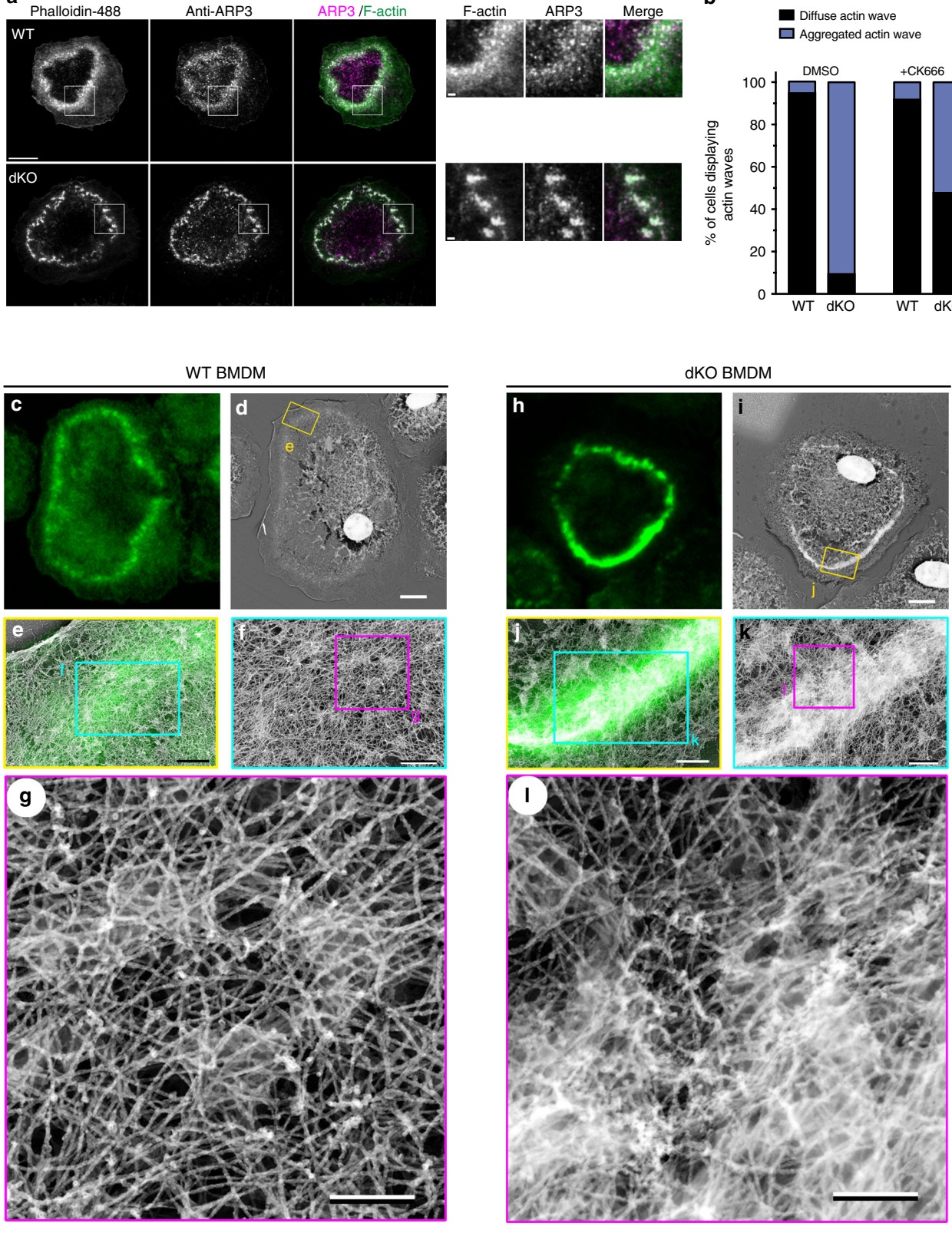

to the actin clumps of the dKO macrophages (Fig. 5a). Treating dKO macrophages with CK-666, an Arp2/3 inhibitor, produced a partial rescue of actin wave morphology (Fig. 5b). Given the abnormal appearance of the phagocytic F-actin in the absence of myo1e/f, we were interested in examining its supramolecular organization. We therefore performed correlative platinum replica electron microscopy (PREM) of WT and dKO macrophages undergoing frustrated phagocytosis. In control cells, the actin wave observed by PREM appeared as small clusters of branched and unbranched actin (Fig. 5c–g, Supplementary Fig. 9a and Supplementary Movie 12). In the case of dKO cells, PREM revealed a much greater density of branched and unbranched

**Fig. 5** Actin waves in dKO BMDM have higher density of branched actin filaments. **a** Immunostaining of Arp2/3 complex in WT and dKO BMDM during frustrated phagocytosis. Cells were counter-stained with fluorescently labeled phalloidin. Zoom of boxed regions shown on the right. Scale bar, 10 µm; zoom panel scale bar, 1 µm. **b** Treating dKO cells with low doses of Arp2/3 inhibitor CK-666 partially rescues actin wave morphology. Graph depicting the mean percentage of cells forming diffuse or aggregated actin waves of WT and dKO BMDM in the presence of 1 µM CK-666. Data from two independent experiments ($n \geq 40$ cells per genotype, judged blindly). **c–g** Correlative confocal and platinum replica EM of actin wave in a representative WT macrophage. **c** Confocal section of a WT cell stained with fluorescently labeled phalloidin for correlative fluorescence image. Scale bar, 5 µm. **d** Platinum replica EM image of the macrophage shown in **c**. Scale bar, 5 µm. **e** Overlay of the enlarged confocal (green) and platinum replica EM (gray) images corresponding to the boxed region in **d**. Scale bar, 0.5 µm. **f, g** Sequential magnifications of boxed regions in **e** and **f**, respectively, showing supramolecular architecture of individual phagocytic adhesions. Scale bars, 0.2 µm. **h–l** Correlative confocal and platinum replica EM of actin wave in a representative dKO macrophage. **h** Confocal section of a dKO cell stained with fluorescently labeled phalloidin for correlative fluorescence image. Scale bars, 5 µm. **i** Platinum replica EM image of the macrophage shown in **h**. Scale bars, 5 µm. **j** Overlay of the enlarged confocal (green) and platinum replica EM (gray) images corresponding to the boxed region in **i**. Scale bars, 0.5 µm. **k, l** Sequential magnifications of boxed regions in **j** and **k**, respectively, showing unbridled actin polymerization. In **l**, image contrast is changed by adjusting gamma. Scale bars, 0.2 µm. See also Supplementary Fig. 9 and Supplementary Movie 12

---

actin, elevated to a significant height above the central surface (Fig. 5h–l, Supplementary Fig. 9b and Supplementary Movie 12). We observed no apparent difference in the relative abundance of branched vs. linear actin within the actin waves of WT and dKO cells. Taken together, these results suggest myo1e/f are constraining Arp2/3-mediated actin polymerization within the phagocytic cup.

**Myo1e/f regulate actin dynamics at phagocytic adhesion sites.** Using live-cell imaging of frustrated phagocytosis by TIRFM, we were able to observe distinctly different actin dynamics between WT and dKO cells. The actin wave of WT cells moved in a sweeping fashion, with new adhesions forming quickly. These adhesions remained uniformly small and were rapidly disassembled during wave movement or collapse (Fig. 6a, b, Supplementary Movie 13). Conversely, the actin adhesion sites in the dKO macrophages moved much more slowly, with distinct fusion and fission events between adhesions leading to their clumped appearance (Fig. 6a, b, Supplementary Movie 13). This was particularly clear during wave stabilization or disassembly in which the formation of giant adhesions seemed to impair wave progression (Fig. 6b). To measure actin turnover rates within the phagocytic adhesions, we used fluorescence recovery after photobleaching (FRAP) on WT and dKO macrophages expressing EGFP-actin (Fig. 6c–f). We observed that in the presence of myo1e/f, actin was more dynamic, recovering to 86% of pre-bleached values. The immobile fraction of actin in the dKO cells was almost three times higher than that of WT macrophages (Fig. 6f). This indicated that the clumped actin in dKO cells was largely composed of stable actin filaments.

To better understand how the actin wave dynamics of the dKO macrophages relates to the slower rate of phagocytic cup closure we observed in our bulk assay, we tested whether increased fluorescence intensity of actin at individual adhesions was associated with reduced actin wave speed in dKO macrophages by quantifying both local boundary speed of the wave edge and the adjacent actin fluorescence intensity (Fig. 6g, Supplementary Movie 14). To extract these data, the exterior of the actin wave was tracked throughout the time-lapse with local wave boundary speed and mean actin intensity measured at each angle of the circular actin wave. Actin wave protrusion occurred if the radial distance change between consecutive frames was positive while a negative number indicated wave retraction. The mean actin intensity with corresponding actin wave boundary speeds, indicated on the X-axis, is shown in Fig. 6h. The majority of our measurements, with high actin adhesion intensities, corresponded to a stalled actin wave (Fig. 6h, see also Supplementary Fig. 10). Meanwhile, lower actin intensities were associated with faster protrusive wave speeds. This analysis illustrates how the lack of myo1e/f leads to the formation of overgrown adhesive patches

with low actin turnover, which effectively impair actin wave motility. Using actin wave speed in this assay to approximate adhesion turnover along the target within the phagocytic cup, we propose this sluggish behavior of the dKO cells slows down phagocytic cup closure. This would explain the actin-dense phagocytic cups of dKO macrophages (Fig. 4h) and the observed delay in bead engulfment (Fig. 1h).

**Myo1e/f promote membrane lifting around phagocytic adhesions.** To address how myo1e/f restrict adhesion size to promote faster turnover and efficient phagocytosis, we focused our attention on the core function of myosin-I: their mechanical role at the interface between membrane and the actin cytoskeleton[52,53]. As an actin-membrane linker, myosin-I can potentially regulate cortical tension (a mostly cytoskeleton-dependent property) as well as membrane tension, which is regulated in part by proteins that link the plasma membrane to the underlying actin cortex[54]. To test this, we used atomic force microscopy (AFM) to probe the cell stiffness of WT and dKO macrophages. These AFM measurements revealed dKO macrophages to be significantly softer (Fig. 7a). As cell stiffness encompasses both cortical tension and membrane tension, we set out to evaluate the contribution of myo1e/f to membrane tension with the tether-pulling assay using optical tweezers[55]. The tether force of dKO macrophages was significantly lower than that of WT macrophages, identifying myo1e/f as contributors to membrane tension in macrophages (Fig. 7b). We previously reported that macrophages undergoing phagocytosis experienced ~30% increase in membrane tension[9]. Therefore, we decided to measure membrane tension while BMDMs were undergoing phagocytosis. Using the optical trap, IgG-opsonized beads were placed in contact with cells to initiate phagocytosis, followed by a laser trap tether force measurement close to the phagocytic cup using another smaller bead (Fig. 7c). Using WT macrophages, we also observed an increase in membrane tension during phagocytosis compared to resting cells. However, no such increase occurred in dKO cells (Fig. 7b), suggesting myo1e/f actively control membrane tension over the course of internalization.

During phagocytosis, we observed myo1e/f at two main locations: the leading edge of the phagocytic cup and phagocytic adhesion sites within the cup (Fig. 3h). The force of actin polymerization during leading edge protrusion has been described as a key contributor to membrane tension regulation in multiple cell systems including macrophages[9,56,57]. However, we detected no difference in leading edge velocity during frustrated phagocytosis in WT and dKO macrophages (Supplementary Fig. 4h).

We next questioned whether myo1e/f might be regulating membrane tension locally at FcR adhesion sites, and, at the same time, restricting adhesion expansion and fusion. While imaging

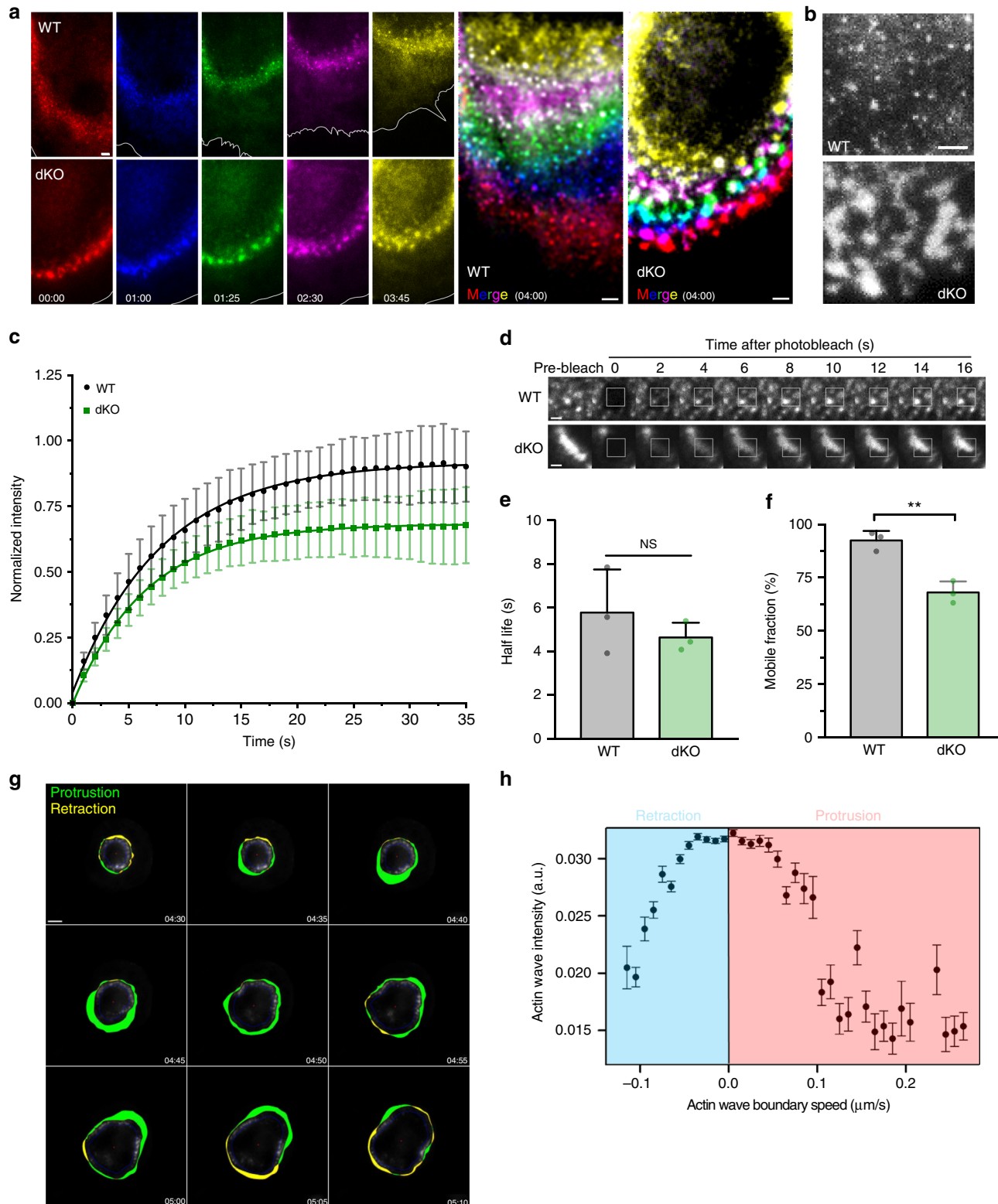

actin dynamics during frustrated phagocytosis by TIRFM, we noticed an interesting characteristic of phagocytic adhesions. In RAW cells co-transfected with a myo1e/f construct and membrane label, FcR adhesion sites were often surrounded by a circular area devoid of the membrane fluorescent signal, suggesting that the ventral surface of the cell in this circular area was located above the TIRF plane of excitation (Fig. 7d). Imaging the membrane by both TIRFM and epifluorescence proved this

lifting to be specific to the ventral cell surface. The shape and area of membrane lifting were variable, ranging from 3 to 10 μm², and while most FcR-myo1 puncta were located centrally within the raised membrane areas, some puncta were located off-center (Fig. 7e). Membrane lifting was also observed in BMDMs (Fig. 7f, Supplementary Movie 15), and we hypothesized that it could be the result of a myo1e/f-dependent membrane lifting away from the substrate around sites of FcR adhesions. In agreement with

**Fig. 6** Myo1e/f regulate actin dynamics at phagocytic adhesion sites. **a** Actin waves dynamics in WT and dKO macrophages transfected with EGFP-actin performing frustrated phagocytosis and imaged by TIRFM. Montage has been color-coded with respect to time (minutes:seconds) to display adhesion turnover. The white line marks the cell's edge. The final images at the right (Merge) are maximum intensity projections. Scale bars, 2 μm. See also Supplementary Movie 13. **b** Gray scale zoom of maximum intensity projections in **a** to show distinct punctate structure of WT adhesions compared to fused/enlarged adhesions in dKO macrophage. Scale bar, 2 μm. **c** Graph of FRAP curves (mean ± SD). WT and dKO BMDM transfected with EGFP-actin were subjected to FRAP analysis to measure actin turnover during frustrated phagocytosis. The resulting data was corrected for photobleaching, normalized to the pre-bleached images, then fit using a single exponential function. Data pooled from three independent experiments ($n = 38$ WT and 36 dKO cells). **d** Representative actin wave recovery assessed by FRAP. White box indicates bleached region. Scale bar, 1 μm. **e** Graph of half-life of recovery (mean ± SD). Data from three independent experiments, ($p = 0.39$, unpaired $t$-test). **f** Graph of mobile fraction (mean ± SD). Data from three independent experiments, $p = 0.0035$, unpaired $t$-test. **g** Time-lapse montage of tracking analysis to characterize dKO macrophage actin wave dynamics. The border of the actin wave was tracked for protrusion (green)/retraction (yellow) speed and the inner blue line marks the inner boundary for actin quantification. Time, minutes:seconds. Scale bar, 5 μm. See also Supplementary Movie 14. **h** Increased actin wave intensity correlated with stalled actin wave. Graph (mean ± SD) showing the distribution of intensities of actin adhesions plotted against actin wave boundary speed during frustrated phagocytosis in a dKO BMDM. Error bars represent variability between individual boundary speeds measurements. See also Supplementary Figure 10

this hypothesis, no membrane lifting was observed near the aggregated actin adhesions in the dKO macrophages (Fig. 7g, Supplementary Movie 16). By averaging the TIRFM intensities of the actin and membrane label around individual adhesions in multiple cells, we found that membrane lifting was consistently observed at WT but not dKO adhesions (Fig. 7h).

In light of these results and in combination with our AFM and tether force measurements, we propose that myo1e/f are tethering the plasma membrane to actin at individual adhesion sites, restricting adhesion expansion by locally lifting the membrane around adhesions, and thus regulating membrane tension (model, Fig. 7i). When individual adhesions fuse together in the absence of myosin-I-mediated membrane lifting, this results in the formation of large actin arrays with low turnover, which ultimately slows down the closure of the phagocytic cup.

## Discussion
Here we show that both myo1e/f are required for efficient FcR-mediated phagocytosis. Compared to other proteins involved in phagocytic ingestion, myo1e/f are uniquely localized at the tip of the phagocytic cup, preceding actin polymerization. We observed myo1e/f between F-actin and membrane at FcR adhesions, with myo1e/f being located slightly more ventrally than actin, in both the 2D frustrated phagocytosis assay and inside the 3D phagocytic cup. Using deletional analysis, we discovered that localization of myo1e/f to the actin wave (or phagocytic cup) depended partly on the TH2 domain in the tail and a functional motor domain. This demonstrates a specific role for long-tailed myosin-Is in phagocytosis, as short-tailed myosin-Is do not contain this tail region. These findings parallel myosin-I behavior reported in *Dictyostelium*, in which planar actin waves are also used as a model for the phagocytic cup[58,59]. The TH2 domain of long-tailed myosin-Is contains an ATP-insensitive actin-binding site[60–62] as well as a large number of basic amino acids that could bind to membrane phospholipids[15]. Thus, it appears that myo1e/f recruitment to the phagocytic cup may depend in part on actin binding via the motor domain and in part on the interactions of the TH2 domain with actin filaments or membrane phospholipids.

The process of phagocytosis involves several steps that could potentially rely on myosin-I functions. Extension of pseudopods during initial cup formation, focal exocytosis to increase cup surface area, and contraction of the actin filaments within the cup could all be driven by myosin activity. However, we found that none of these events appear to be connected to the activity of myo1e/f. Phagocytic cups still formed in the absence of these myosins, indicating that initial actin assembly and pseudopod extension were not disrupted. Focal exocytosis markers did not

colocalize with myo1e/f, and dKO macrophages did not exhibit defects in membrane dynamics suggesting myo1e/f have no direct role in membrane addition to the cup. Unexpectedly, the loss of myo1e/f led to a marked change in actin dynamics during frustrated phagocytosis, resulting in excessive accumulation of branched actin networks with low turnover at FcR adhesion sites. By measuring membrane tension, we confirmed that myo1e/f function to connect the plasma membrane to the underlying actin cortex in macrophages, and that an increase in membrane tension that is normally induced by phagocytosis is not observed in myo1e/f null cells.

Using TIRF imaging of cells undergoing frustrated phagocytosis, we observed plasma membrane lifting away from the IgG-coated substrate in the areas surrounding FcR- and actin-supported adhesions. In macrophages lacking myo1e/f, this membrane lifting was not observed, suggesting that the loss of membrane-actin tethering alters membrane geometry at phagocytic adhesion sites. Past electron microscopy studies similarly show that in cross-sections of phagocytic cups, cell-target contact sites are not continuous but rather separated by regions where the plasma membrane is lifted away from the surface of the target[63,64]. These electron-dense contact areas were proposed to be the morphological representation of the zipper mechanism. Recently, these adhesions, which resemble the teeth of a zipper in the zippering model, have been shown to be distinct areas of cell-based compression during phagocytosis, with forces on the order of those produced by macrophage podosomes[65,66]. From our observations, we propose that myo1e/f are required to tether plasma membrane around the individual teeth of the zipper away from the target surface. A second hypothetical mode of action could be the formation of podosome-like projections at these teeth. Podosomes are known to exhibit membrane puckering around their actin cores[67,68], which contain myo1e/f in primary macrophages[69]. In the absence of membrane lifting by myo1e/f, flattened plasma membrane under reduced tension facing the phagocytic target enables the formation of dense F-actin mats that turn over more slowly. Membrane tension may affect actin polymerization directly, by generating a force that opposes the growth of actin filaments, or indirectly, by regulating recruitment of actin polymerization regulators. During neutrophil migration, membrane tension provides mechanosensitive feedback that limits or redirects F-actin polymerization[70,71]. Interestingly, in worm sperm cells, reduced membrane tension results in less organized cytoskeletal filaments in lamellipodia and a slower extension speed[72], suggesting that membrane tension may determine cytoskeletal filament organization.

An alternative explanation for the excessive actin polymerization phenotype in the absence of myo1e/f is that these myosins could be necessary for the activity or localization of actin

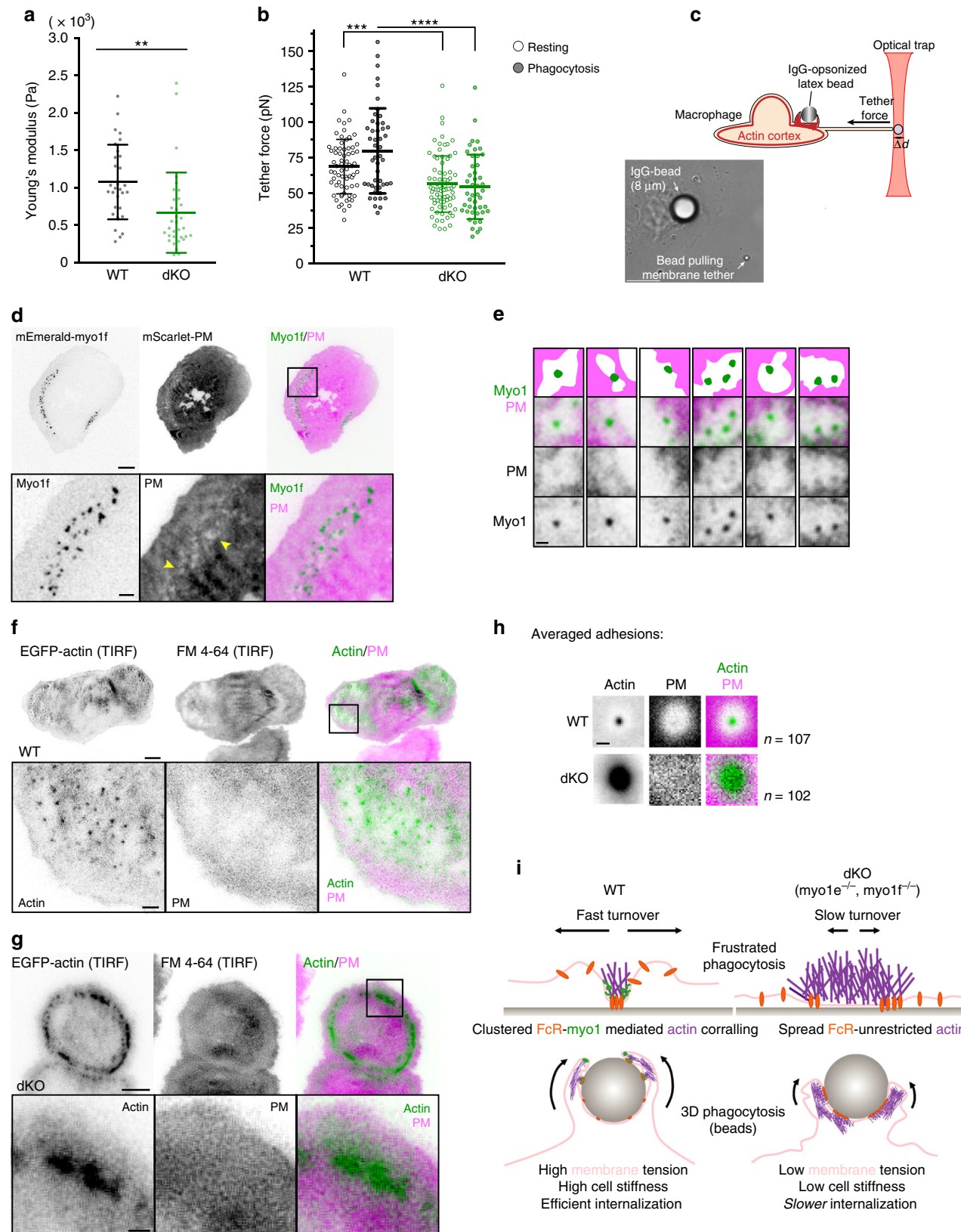

depolymerizing or severing proteins. Using an antibody against capping protein, which has an important role in limiting actin assembly, we could not detect any difference in the ratio of capping protein to F-actin in WT and dKO cells conducting frustrated phagocytosis (Supplementary Fig. 11a-b). Moreover,

myo1e has been reported to bind actin regulatory proteins such as CARMIL[73], and we have found no difference in the recruitment of these proteins during frustrated phagocytosis (Supplementary Fig. 11c-d). Intriguingly, a recent paper has reported that F-actin depolymerization rates increase when actin filaments slide on

**Fig. 7** Myo1e and myo1f promote local membrane lifting around phagocytic adhesion sites. **a** Cytoskeletal tension of WT and dKO BMDM (mean ± SD) measured using atomic force microscopy ($n = 30$ WT and 33 dKO cells, $p = 0.0026$, unpaired $t$-test). **b** Membrane tension (mean ± SD) measured in WT and dKO BMDM while resting (data pooled from two independent experiments, $n = 69$ WT and 75 dKO cells, $p = 0.0003$, unpaired $t$-test) and performing phagocytosis (data pooled from two independent experiments, $n = 48$ WT and 46 dKO cells, ****$p < 0.0001$, unpaired $t$-test). **c** Schematic of membrane tension measurement by the tether-pulling assay during phagocytosis. Beneath, representative bright-field image of assay. Scale bar, 10 μm. **d** Membrane lifting observed around phagocytic adhesions. TIRFM imaging of frustrated phagocytosis in RAW macrophage expressing mEmerald-myo1f and mScarlet-PM (plasma membrane marker) (inverted intensity for myo1f and PM). Yellow arrowheads in the zoomed image of the boxed region denote sites of membrane lifting around phagocytic adhesions. Scale bar, 10 μm; zoom scale bar, 2 μm. **e** Examples of myo1e/f placement at membrane lifting sites. TIRFM images of frustrated phagocytosis in RAW macrophages transfected with membrane marker and myo1e/f (inverted intensity for myo1 and PM). Top row contains schematic representations corresponding to merged panels. Scale bar, 1 μm. **f, g** Membrane lifting around phagocytic adhesions observed in WT (**f**), but not dKO (**g**) BMDM. WT and dKO BMDM with EGFP-actin and stained with membrane label FM 4-64 were imaged by TIRFM during frustrated phagocytosis (inverted intensity for actin and PM). Scale bar, 5 μm; inset scale bar, 2 μm. **h** Inverted fluorescence intensity maps obtained by averaging multiple phagocytic adhesions in WT and dKO BMDM and showing membrane (FM 4-64) relative to F-actin. Data from three independent experiments ($n = 107$ WT and 102 dKO adhesions from at least 15 cells). Scale bar, 1 μm. **i** Graphical model: myo1e/f tether membrane at phagocytic adhesion sites. Without these linkages, local membrane tension is altered, allowing unrestricted actin polymerization causing enlarged phagocytic adhesions. This results in an actin-dense phagocytic cup that completes closure at a slower rate

---

myosin 1b (myo1b)[74]. However, since the kinetics of the myo1b motor domain are distinct from that of long-tailed class 1 myosins, this feature is unlikely to be generalizable to myo1e/f.

Overall, our findings emphasize a heretofore unappreciated component of macrophage phagocytic cup closure: dynamic adhesions at the bead-membrane interface, which closely couple actin polymerization and increased membrane tension for target engulfment. While myo1e/f in macrophages regulate membrane tension globally, their concentrated action within the phagocytic cup appears to control the size and consequently dynamics of phagocytic adhesions. Interestingly, our observations appear to provide a mechanistic, molecular basis for the enveloping embrace model, which suggests phagocytes use myosin-I-based adhesions that couple the actin cytoskeleton to the plasma membrane to effectively anchor the target to the cell allowing F-actin polymerization within the cup to push forward and finish internalization[75]. Furthermore, as phagocytosis is known to be a mechanosensitive process[76], these dynamic phagocytic adhesions that rely on myosin-I-regulated membrane tension may possess mechanosensory properties, allowing cells to probe mechanical parameters of phagocytic targets. This possibility represents an exciting topic for future studies.

## Methods

**Mice.** Myo1e$^{-/-}$ mice[23] and myo1f$^{-/-}$ mice[24] were crossed to create myo1e$^{-/-}$; myo1f$^{-/-}$ dKO mice. All mice used in this study were maintained on a C57BL/6 background. All procedures utilizing mice were performed according to animal protocols approved by the IACUC of SUNY Upstate Medical University and in compliance with all applicable ethical regulations. Both male and female mice were used; for each individual experiment utilizing BMDM, bone marrow preparations from age- and sex-matched mice were used.

**Cell culture.** RAW264.7 (ATCC; male murine cells) were cultured in Dulbecco's Modified Eagle Medium (DMEM), high glucose, containing 10% FBS and 1% antibiotic-antimycotic (Gibco) at 37 °C with 5% $CO_2$. All transfections of RAW cells and BMDM were accomplished by electroporation (Neon) using manufacturer's instructions.

**Bone marrow isolation and primary macrophage culture.** Following euthanasia, femurs and tibias of mice were removed and flushed with Dulbecco's Modified Eagle Medium (DMEM) containing 10% FBS and 1% antibiotic-antimycotic (Gibco). Red blood cells were lysed using ACK buffer (0.15 M NH₄Cl). Bone marrow progenitor cells were recovered by centrifugation (250×g, 5 min, 4 °C), washed once with sterile PBS and plated on tissue culture dishes in a 37 °C incubator with 5% $CO_2$. The next day, non-adherent cells were moved to bacteriological (non-tissue-culture treated) Petri dishes and differentiated over 1 week in DMEM, high glucose with 20% L929-cell conditioned media (v/v), 10% FBS, and 1% antibiotic-antimycotic, with fresh medium given on day 3 or 4. Cells were confirmed to be macrophages (F4/80$^+$ CD11b$^+$) by flow cytometry. All experiments were done within five days post-differentiation.

**Chemicals and drugs.** Latrunculin A, CK-666, and LY294002 were purchased from EMD Millipore. Jasplakinolide, Concanavalin A, and unlabeled phalloidin were purchased from Sigma. Alexa Fluor-488 or Alexa Fluor-568 conjugated phalloidin were purchased from Life Technologies and FM 1-43 or FM 4-64 were purchased from Invitrogen. Human fibronectin was purchased from Corning and poly-L-lysine (50–70 kDa) from Sigma was prepared as a 10 mg/mL stock.

**Antibodies.** The following antibodies were used in this study: rabbit anti-myo1e has been previously described[77]; myo1f (Santa Cruz, B-5, #376534); capping protein (Millipore, #AB6017); Arp3 (Millipore, clone 13C9, #MABT95); BSA (Sigma, clone 3H6, #SAB5300158); rat anti-mouse CD16/32 (BD, #553141); AffiniPure mouse anti-rat (Jackson Labs, 212-005-082); the following antibodies were purchased from Cell Signaling Technologies: pSyk (#2701), Syk (#13198), pAkt (#4060), Akt (#4691), pERK (#4370), ERK (#9102); fluorescent secondary antibodies against mouse or rabbit (Life Technologies). CARMIL1 antibody (WU-C101) was a gift from John Cooper, Washington University in St. Louis (St. Louis, MO).

**Constructs.** Human myosin 1e and truncated constructs tagged with EGFP have been previously described[78]. Human myosin 1f (NM_012335.3) cDNA from the Mammalian Gene Consortium was amplified and cloned into pEGFP-C1 (Clontech). Both myosins were also cloned into mEmerald-C1 and tdTomato-C1 (gifts from Michael Davidson; Addgene plasmids #53975 and 54653) and mScarlet-C1. The following constructs were purchased from Addgene: EGFP-PLCδ-PH, EGFP-AKT-PH, and EGFP-PKCδ-C1 (gifts from Tobias Meyer, plasmids #211789, 21218, and 21216), mScarlet-PM (gift from Dorus Gadella, #98821), EGFP-VAMP3 (a gift from Thierry Galli, #42310), Ruby-Lifeact, mEGFP-Lifeact, mEmerald-Lifeact and EGFP-actin (gifts from Michael Davidson, #54674, #54610, #54148, and #56421). EGFP-FcγRIIA was a gift from Sergio Grinstein (Hospital for Sick Kids, Toronto, ON), and EGFP-TAPP1 was a gift from Aaron Marshall (University of Manitoba, Winnipeg, MB). YFP-myo1c and EGFP-myo1g were gifts from Matt Tyska (Vanderbilt University, Nashville, TN) that were cloned into mEmerald-C1. All primers used for construct generation are listed in Supplementary Table 1.

**Immunostaining.** Cells were fixed using fresh 4% paraformaldehyde/PBS for 15 min. After washing away fixative, cells were permeabilized in 0.1% Triton X-100/PBS for 3 min. Cells were blocked for 30 min at room temperature with 5% normal goat serum/3% BSA dissolved in PBS with 0.05% Tween-20. Cells were exposed to primary antibodies at the appropriate dilutions for 1 h at room temperature. Cells were then washed three times for 5 min with PBS/0.05% Tween-20. Secondary antibodies and fluorescent phalloidin were then applied for 30 min at room temperature. Cells were then washed again for three times, 5 min each before mounting with Prolong Diamond Antifade Mountant (Invitrogen). Primary antibodies were used at the following concentrations: myo1e (1:600), ARP3 (1:100), capping protein (1:100), CARMIL1 (1:100).

**FcR stimulation.** Fc receptor stimulation[25,79] was conducted using BMDM that were plated at $6 \times 10^6$ cells per 10 cm petri dish. Cells were serum-starved for 6 h and then incubated with 10 μg/mL rat anti-mouse CD16/32 antibody (2.4G) (diluted 1:50) in cold serum-free media for 40 min at 4 °C to allow binding without receptor internalization. Cells were then quickly washed 2× with ice-cold PBS to remove unbound antibody and exposed to warm serum-free media containing 15 μg/mL anti-rat antibody (diluted 1:160) to bind the anti-CD16/32 antibody and incubated at 37 °C to initiate antibody crosslinking and resulting downstream signaling. At the specified time points, crosslinking media was quickly removed

and the cells were quickly washed 1× with cold PBS and processed for western blot analysis. The 0 time point control dish was left at 4 °C.

**Western blotting**. Cells were washed 1× in ice-cold PBS and harvested by scraping in NP-40 lysis buffer (1% NP-40, 150 mM NaCl, 50 mM Tris-HCl, 10 mM NaF) with phosphatase and protease inhibitors (Roche). Cells were rotated at 4 °C for 25 min and then pelleted at 16,000×g for 15 min at 4 °C. Supernatant was then removed and boiled with Laemmli sample buffer, and separated on 10–20% gradient SDS-PAGE gel, followed by transfer to PDVF. Membranes were blocked in 5% milk or 3% BSA in TBST for 1 h at room temperature. Primary antibodies were diluted (myo1e, 1:6000; myo1f, Syk, pSyk, Akt, pAkt, ERK, pERK, 1:1000) in 5% milk or 3% BSA in TBST and incubated with the membrane overnight at 4 °C. The next day, the membrane was washed 3× for 5 min in TBST. HRP-conjugated secondary antibodies were diluted in 5% milk or 3% BSA in TBST and exposed to membranes for 1 h at room temperature. Chemiluminescence was detected using WesternBright Quantum (Advansta) and imaged on a Biorad ChemiDoc imaging system. Immunoblots were stripped using mild stripping buffer (1.5% glycine (w/v), 0.1% SDS (w/v), 1% Tween-20 (v/v), pH 2.2) and reprobed when appropriate. Uncropped blot and gel images are included in the Data Source file.

**Phagocytosis assay**. Polystyrene beads (PolySciences Inc., 2, 6, or 8 μm) were washed three times in sterile PBS and opsonized overnight at 4°C in 3 mg/mL mouse IgG (Invitrogen). To remove excess antibody, beads were washed three times with PBS and resuspended in sterile PBS. Beads were applied to macrophages in a 12-well plate at an estimated ratio of 10:1. To synchronize phagocytosis, the plate was spun at 300 × g for 2 min at 4 °C. Cells were incubated at 37 °C to initiate phagocytosis. To stop phagocytosis, cells were washed three times with ice-cold PBS to remove unbound beads and fixed with 4% PFA/PBS for 15 min. Cells were then washed and stained with goat anti-mouse-Alexa Fluor-568 antibodies for visualization of un-internalized beads for 30 min. Cells were then washed with PBS (3 × 5 min) and permeabilized with 0.025% Triton X-100/PBS, then stained with Alexa Fluor 488-conjugated phalloidin, followed lastly by DAPI (NucBlue Fixed Cell ReadyProbe Reagent, Invitrogen). Coverslips were then mounted using Prolong Diamond Antifade Mountant. For phagocytic internalization/association quantification, 15–30 fields of view per genotype per time point were imaged by spinning disk confocal microscopy using ×20 magnification. Internalized beads were visualized using the bright-field channel. Quantification resulted in 600–1400 cells per genotype being analyzed in total from all time points in one independent experiment.

**Frustrated phagocytosis assay**. Protocol for frustrated phagocytosis assays was adapted from a previous study[32]. In brief, glass coverslips were acid-washed with 20% nitric acid. They were then coated in 1 mg/mL BSA/PBS for 1 h at 37 °C, followed by incubation with 10 μg/mL mouse anti-BSA antibody (1:100 dilution) for 1 h at 37 °C. Coverslips were washed 3× in PBS before use. For live-cell imaging, coverslips were affixed to a custom imaging chamber with vacuum grease (Dow Corning). Prior to the assay, cells were serum-starved in 1× Ringer's buffer (150 mM NaCl, 5 mM KCl, 1 mM CaCl₂, 1 mM MgCl₂, 20 mM HEPES and 2 g/L glucose, pH 7.4) for 20 min. Cells were introduced to the mounted chamber and imaged spreading in 1× Ringer's buffer at 37 °C. In the case of drug studies, cells were exposed to the drug while in suspension and carried out frustrated phagocytosis in the same dilution of the drug.

**Flow cytometry**. Macrophages were pelleted (250×g, 5 min, 4 °C, 1 × 10⁶ cells/ tube) and resuspended in fresh FACS buffer (5% FBS/0.1% NaN₃/PBS) and blocked with rat anti-mouse CD16/32 (Biolegend, clone 93, #101301) for 30 min on ice. Cells were then incubated with FITC-F4/80 (Biolegend, clone BM8, #123107) diluted 1:100 and APC-CD11b (BD Biosciences, clone M1/70, #553312) diluted 1:250 for 50 min in the dark. Cells were washed twice and resuspended in 0.2 mL FACS buffer for immediate analysis with a BD LSRII flow cytometer and BD FACSDiva program. Fc receptors were quantified using APC-CD16/32 (BD, clone 2.4G2, #558636) diluted 1:200. Data were processed using FlowJo software.

**Platinum replica electron microscopy**. For correlative light and platinum replica electron microscopy, cells were allowed to spread on IgG-coated glass pre-coverslips coated with a thin layer of gold through a locator grid (400 mesh, Ted Pella, Inc., Redding, CA). The gold layer provided the coverslips with a finder grating recognizable by both light and electron microscopy[80]. The cells were extracted for 3 min with extraction solution containing 1% Triton X-100, 2% PEG (MW 35,000), 4 μM unlabeled phalloidin in M buffer (50 mM imidazole, pH 6.8, 50 mM KCl, 0.5 mM MgCl₂, 1 mM EGTA, and 0.1 mM EDTA). Samples were triple washed in M buffer with 4 μM unlabeled phalloidin and fixed in 2% glutaraldehyde in PBS buffer for 20 min. Then samples were triple washed in PBS and quenched with 5 mg/mL NaBH₄ in PBS for 10 min and again triple washed in PBS. Next, cells were stained with Alexa Fluor 488-phalloidin and imaged by spinning disk confocal microscopy. Samples for correlative platinum replica electron microscopy were processed as described previously[81,82]. Samples were analyzed using JEM 1011 transmission electron microscope (JEOL USA, Peabody, MA)

operated at 100 kV. Images were captured by ORIUS 832.10 W CCD camera (Gatan, Warrendale, PA) and presented in inverted contrast.

**Traction force microscopy (TFM)**. Linearly elastic polyacrylamide gels with a shear modulus of 1.5 kPa were prepared using published protocols[83], with a final mixture of 7.5% acrylamide/0.05% bis-acrylamide solutions (Bio-Rad). 40 nm dark-red fluorescent beads (Invitrogen) were included in polymerization mixture at 1:100 dilution. 6 μL of the PAA solution was pipetted onto a hydrophobic glass slide, covered with a silanized coverslip (No. 1.5, Electron Microscopy Sciences) and allowed to polymerize for 30 min. The gels were removed and washed in PBS. BSA (Fisher Bioreagents) was coupled to the surface of the gel using the photo-activatable crosslinker sulfo-SANPAH (Thermo Scientific). Gels were incubated in a 2 mg/mL solution of sulfo-SANPAH and exposed for 5 min at 5 W in a UV-crosslinker (Analytik-Jena), rinsed with water, and incubated inverted on parafilm in a 1 mg/mL solution of BSA for 1 h at room temperature. The gels were washed several times in PBS, and further incubated with a mouse anti-BSA antibody (Sigma) at 10 μg/mL for 1 h, and finally washed thoroughly with PBS. Cells were imaged on a Nikon Ti-E Microscope with Andor DragonFly Spinning Disk Confocal system using a ×60 (1.49 N.A.) TIRF Objective. Images were recorded on a Zyla 4.2 sCMOS camera. Cells were maintained at 37 °C and 5% CO₂ in an environmental chamber (Oko Labs). All hardware was controlled using Andor's iQ software. Approximately 50,000 cells were added to the gels immediately prior to imaging. Traction forces were calculated as previously described[84]. Briefly, bead images in a time series were first registered using a region devoid of cells. Bead displacement in the substrate was determined using PIV software (OpenPIV; MATLAB), which compared each bead image to a reference image before the cell had attached to the substrate. Each window was 3.46 μm × 3.46 μm in size with a center to center distance of 1.73 μm. Displacement vectors were filtered and interpolated using the Kriging interpolation method. Traction stresses were reconstructed via Fourier Transform Traction Cytometry, with a regularization parameter chosen by minimizing the L2 curve. The strain energy was calculated as one half the integral of the traction stress field dotted into the displacement field[85]. Spread areas were calculated using DIC images traced by hand.

**TIRF microscopy**. TIRF imaging was performed on multiple systems: (Supplementary Fig. 6c) an Olympus inverted microscope equipped with an iLas2 targeted laser illuminator (Roper Scientific) using both a ×63 and ×100 objective (1.49 N.A.). Fluorescence was spectrally filtered and collected using a pair of Evolve EMCCD cameras (Photometrics) for red and green emission. (Fig. 3e and Supplementary Fig. 5c, d, 6d) a Nikon Eclipse TE2000-E2 multimode TIRF microscope equipped with PRIME-95B cMOS camera (Photometrics), CFI Apo ×100 (1.49 N. A.) oil TIRF objective (Nikon), OKO Labs temperature-controlled microscope enclosure (OKO Labs), LUNA-4 solid state laser (Nikon). All other images were collected on: True MultiColor Laser TIRF Leica AM TIRF MC equipped with an Andor DU-885K-CSO-#VP camera and a ×63 (1.47 N.A.) oil CORR TIRF objective. For phagocytic adhesions averages of actin and membrane, 4 μm × 4 μm squares were drawn around single adhesions of WT and dKO BMDM, transfected with EGFP-actin and stained with FM 4-64 imaged by TIRF. These images were then averaged using ImageJ/Fiji[86]. Over 100 adhesions were measured in at least 10 cells per genotype from three independent experiments.

**Confocal microscopy**. Images were taken using a PerkinElmer UltraView VoX Spinning Disc Confocal system mounted on a Nikon Eclipse Ti-E microscope equipped with a Hamamatsu C9100-50 EMCCD camera, a 100 × (1.4 N.A.) PlanApo objective, and controlled by Volocity software.

**Structured illumination microscopy**. SIM images were acquired using Nikon N-SIM E microscopy system based on the Eclipse Ti research inverted microscope with CFI Apo TIRF SR ×100 (1.49 N.A.) objective and Hamamatsu ORCA-Flash 4.0 V2 camera. Actin adhesion height and area were measured using Imaris software (v. 9.2.1, Bitplane Inc.).

**Lattice light sheet microscopy**. The lattice light sheet microscope[87] was developed by E. Betzig and operated/maintained in the Advanced Imaging Center at the Howard Hughes Medical Institute Janelia Research Campus (Ashburn, VA). 488, 560, or 642 nm diode lasers (MPB Communications) were operated between 40 and 60 mW initial power, with 20–50% acousto-optic tunable filter (AOTF) transmittance. The microscope was equipped with a Special Optics 0.65 NA/ 3.75 mm water dipping lens, excitation objective and a Nikon CFI Apo LWD 25 × 1.1 NA water dipping collection objective, which used a 500 mm focal length tube lens. Live cells were imaged in a 37 °C-heated, water-coupled bath in FluoroBrite medium (Thermo Scientific) with 0–5% FBS and Pen/Strep. Images were acquired with a Hamamatsu Orca Flash 4.0 V2 sCMOS cameras in custom-written LabView Software. Post-image deskewing and deconvolution was performed using HHMI Janelia custom software and 10 iterations of the Richardson-Lucy algorithm.

**Atomic force microscopy**. AFM indentation was carried out using JPK Nano-Wizard3 mounted on an Olympus inverted microscope. The protocol was adapted

from a previous study[88]. A modified AFM tip (NovaScan, USA) attached with 10 μm diameter bead was used to indent the center of the cell. The spring constant of the AFM tip cantilever is ~0.03 N/m. AFM indentation loading rate is 0.5 Hz with a ramp size of 3 μm. AFM indentation force was set at a threshold of 2 nN. The data points below 0.5 μm indentation depth were used to calculate Young's modulus to ensure small deformation and minimize substrate contributions[88]. The Hertz model is shown below:

$$F = \frac{4}{3}\frac{E}{(1-\nu^2)}\sqrt{R\delta^3} \qquad (1)$$

where $F$ is the indentation force, $E$ is the Young's modulus to be determined, $\nu$ is the Poisson's ratio, $R$ is the radius of the spherical bead, and $\delta$ is indentation depth. The cell was assumed incompressible and a Poisson's ratio of 0.5 was used.

**Phagocytic cup F-actin quantification**. Phagocytic cups of WT and dKO BMDM, fixed and stained with fluorescently labeled phalloidin, were imaged using a ×100 objective by spinning disk microscopy with 0.3 μm z-steps. Images were reconstructed using Imaris software and the average fluorescence intensity per voxel and integrated intensity were measured for an ROI enclosing the cup. Over 150 cups per sample were analyzed in three independent experiments.

**FM 1-43 quantification**. The protocol for FM 1-43 experiments was adapted from a previous study[9]. Serum-starved resuspended macrophages were exposed to 10 μg/mL FM 1-43 before being added to an imaging chamber containing background 5 μg/mL FM 1-43 in 1× Ringer's buffer. As cells spread, DIC and FITC images were collected at 15 s intervals using a 40× (1.35 N.A.) water objective mounted on a DeltaVision microscope (Olympus IX70; Applied Precision) equipped with a CoolSNAP HQ camera (Photometrics) and SoftWoRx software. As FM 1-43 has a quantum yield 40× higher in lipid membranes, total FM 1-43 intensity from a cell at a given time point is linearly related to the sum of the initial plasma membrane area and total net membrane exocytosed. Intensities were extracted from images and background corrected using ImageJ before being normalized to the initial pre-spread intensities.

**Actin wave intensity and boundary speed quantification**. To analyze the actin wave boundary speed, we wrote a custom macro for Fiji, which is available upon request. Signal background was subtracted and the actin wave was manually outlined in all frames. Once the cell center was identified, for each angle between 0 and 359°, the actin wave boundary speed was computed by measuring the distance between the actin wave boundaries in two consecutive frames. Speed is defined positive when the actin wave boundary expands and negative when it contracts. The corresponding mean actin intensity of the wave for the same angles was computed by averaging the signal intensity from the actin wave boundary to 30 pixels inward (~4 μm). For comparisons between cells, actin intensity measurements were normalized to 1 at null speed. Results were analyzed and plotted with R[89] and the ggplot2[90] R package.

**FRAP analysis**. Fluorescence recovery after photobleaching (FRAP) was performed using a PerkinElmer UltraView VoX Spinning Disk Confocal system equipped with the Photokinesis module. Photobleaching using full power of a 488 nm argon laser was performed by selecting a square ROI. Post-bleach images were collected at 1 s intervals. Changes in ROI fluorescence intensity were measured over time using ImageJ and corrected using the background intensity and a control region of interest to account for any acquisition bleaching and normalized to pre-bleach and post-bleach intensity values. The best fit curve for fluorescence recovery was obtained using Prism Software. The following single exponential equation was used:

$$y = y_0 + a \cdot (1 - e^{-bt}) \qquad (2)$$

where $x$ is seconds. The half time of recovery was determined using $b$ from the previous equation, where

$$t_{1/2} = \ln 0.5 / -b \qquad (3)$$

Percentage of recovery (mobile fraction) was calculated using formula: $X_m = F_\infty/F_i$. Where $F_i$ denotes the average fluorescence intensity before photobleaching for each normalized curve, $F_\infty$ refers to the average fluorescence intensity derived from the plateau for each normalized curve.

**Membrane tension measurements**. Membrane tension was measured as previously described[9]. In summary, an optical tweezer was generated on a Nikon A1-R microscope by focusing a 5.0-W, 1064-nm laser through a 100× (1.3 N.A.) objective (Nikon). 1 μm polystyrene beads (Polysciences Inc.) coated in Con A were used to pull tethers from macrophages. Bright-field images were acquired using an Andor camera. The trap strength was calibrated with the help of a previously described method[91], by observing Brownian motion of trapped beads with an exposure time of 0.6 ms to minimize motion blur. The measured bead displacement was tracked using ImageJ and converted into measured force.

**Statistical analysis**. Comparisons between WT and dKO BMDM were carried out using an unpaired two-tailed $t$-test for independent samples, with differences between genotypes considered statistically significant at $p < 0.05$. If the standard deviation of the two samples differed markedly, a Welch's $t$-test was used. For multiple comparisons, data were analyzed using a one-way ANOVA with Tukey's post-hoc test, with statistical significance set at $p$-value < 0.05. Statistical analyses and graphing was performed by GraphPad Prism software.

**Reporting summary**. Further information on experimental design is available in the Nature Research Reporting Summary linked to this article.

## Data availability

Data supporting the findings of this manuscript are available from the corresponding authors upon reasonable request. A reporting summary for this Article is available as a Supplementary Information file. The source data underlying Figs. 1g, h, j, k, 2b–d, 4b, d, e, g, h, 5b, 6c, e, f, h, 7a, b and Supplementary Figs. 2b, d, 3b, c, 4b, c, f–h, 7b, c, 8a, 10, 11b, d are provided as a Source Data file.

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

## Acknowledgements

This work was supported by the NSF (#1515223) and AHA (18PRE34070066) grants to S.R.B., the National Institute of Diabetes and Digestive and Kidney Diseases of the NIH under Award R01DK083345 to M.K., the National Institute of General Medical Sciences grant R01GM095977 to T.S., the Mechanobiology Institute of Singapore grant WBS R-714-016-007-271, and the Italian Association for Cancer Research (AIRC), Investigator Grant (IG) 20716 to N.C.G. A travel award grant from the Boehringer Ingelheim Fonds to S.R.B. also made this work possible. We would like to thank Phuson Hulamm and Nicholas Deakin, Ph.D. (Nikon Instruments Inc.) for assistance with SIM image acquisition. LLSM imaging was performed at the Advanced Imaging Center (AIC)—Howard Hughes Medical Institute (HHMI) Janelia Research Campus. We thank Eric Wait and Jesse Aaron of the AIC for helpful discussion. The AIC is jointly funded by the Gordon and Betty Moore Foundation and the Howard Hughes Medical Institute. We gratefully acknowledge technical assistance from Sharon E. Chase.

## Author contributions

S.R.B. designed and performed experiments, analyzed data, and wrote the manuscript. N.S.R. and P.W.O. performed TFM experiments and analyzed TFM data. M.S.S. and T.S. performed platinum replica EM studies. Q.L. assisted with AFM experimentation and analysis. P.M. helped analyze data. J.M.H assisted with LLSM imaging experiments. M.S.M. and R.A.F. generated mouse model. M.K. and N.C.G. designed experiments and wrote the manuscript. All authors reviewed the manuscript prior to publication.

## Additional information

**Competing interests:** The authors declare no competing interests.

