## [Peer Review File · Nature Communications]

Reviewers' Comments:

Reviewer #1:

The manuscript by Barger et al entitled 'Membrane-cytoskeleton mechanical feedback mediated by Myosin-I controls phagocytic efficiency' is a well written and interesting report that addresses for the first time the role of myosins 1e and 1f in the phagocytic cup. The manuscript is rich with convincing data and the overall result should be relevant to wide range of researchers. The authors employed an impressive range of sophisticated techniques (FRAP, AFM, optical tweezers, traction force, FACS, SIM, correlative fluorescence and EM, etc.) to perform elegant experiments in order provide a detailed characterization of myosin 1e/1f. They used a combination of standard phagocytosis and frustrated phagocytosis assays with primary macrophages to achieve their goals.

The work presents compelling data to support a series of claims which together culminate to support the authors' hypothesis that the work in total identifies a novel role for class 1 myosins in coordinated adhesion turnover during phagocytosis.

Their main results include what is effectively a summary of the titles listed throughout the manuscript. I will briefly summarize the most important findings:

1. Myo 1e/f are required for efficient phagocytosis and are co-localized with at the front of the phagocytic cup with actin (actually slightly in front of the actin)
2. Intriguingly myo1e/1f are not required for FcR signaling, target binding or phagocytic cup formation. They are not involved in contraction during phagocytic cup closure nor do they participate in focal exocytosis.
3. Myo1e/1f localize to FcR-actin adhesions during frustrated phagocytosis and that their presence helps prevent the overgrowth of actin potentially by lifting the cell plasma membrane from the surface. Indeed if cells lack myo1e/1f, this results in the formation of disorganized actin-based adhesions and the formation of denser phagocytic cups.

Overall, the work is a very thorough study which is well assembled and reported. There are few instances where clarifications or corrections would improve the document – described below. Further I have one main concern regarding their second claim (in the abstract):

The manuscript's second main claim is that the results 'support a model for a membrane-tension based feedback mechanism for phagocytic cup closure' is less clear. This is my largest critique of the manuscript – that this claim would be better supported by an expanded discussion of their proposed mechanism for how Myo1 regulates membrane tension. Specifically it would be helpful if they layout a step-by-step timeline of events, being sure to include when and how myo1 regulates tension and why that cascades to down-stream process to complete phagosome closure. A schematic would help tremendously. Unlike the rest of the manuscript which reads very smoothly and clearly with good messaging, this other key claim is less well fleshed out. The mere fact that the summary of the key data results above do not directly include tension shows why / how this point might be less clear or not well supported.

Other minor feedback:

In some cases the authors give the number of cells assessed (N) and in other cases they don't. For all data presented, the authors should report the number of cells measured in all reported results, so that readers have an indication of what the actual spread in values are and not just the uncertainty of

the mean. For example, Figure 1h, j, k do not include the number of cells. Similar, when quantifying actin adhesions (area, height), the total number measured should be reported (see Fig 4d-f)

Now some feedback organized around the figures:

Figure 1

- Figure 1b shows a cross section for Fig 1a and says it corresponds to dotted line in Fig 1a. I don't see that line – at least not in my print out
- Fig 1d has reversed colors for myosin and actin relative to many of the other images in the same figure, including 1a-c and 1f; I recommend switching them for consistency and ease of digestion
- Fig 1f – I don't find this figure very informative. It isn't clear that the yellow arrow is pointing to the edge of the phagocytic cup – the entire bead appears to be coated with fluorescence.
- Fig 1h – as stated above, it would be beneficial to have data on the total number of cells measured rather than just the % phagocytic cells. I won't repeat this for every figure.
- Fig 1i – say in the caption what cell type these are BMDM or RAW264.7

Figure 4.

- What is the criteria to distinguish between a diffuse actin wave and aggregated wave? How was the data in Fig 4b evaluated? In Fig 4c, why are the puncta distributed all over the cell rather than in a diffuse ring/wave? Is this expected?
- Why isn't an image shown from the 3DSIM to show the actin adhesion height? Fig 4e? It would be useful to show so the reader has a sense of the quality of the data.
- What kind of resolution does the 3DSim afford in z? It is worth stating this since it is a relatively new technique and it could vary from setup to setup.

Figure 6

Fig6c: is it SEM or St dev shown in the error bars?

It's not clear what is being measured and reported in the plots in Fig 6 h and i. How is each point in Fig 6h determined? Is it some pixel bin located at the cell periphery? Is it an average for the whole cell? Similarly, how was the data averaged for Fig 6i is that one cell or an average of many?

Figure 7

I found Fig 7 hard to interpret. In particular the "donut" of stained membrane around the actin puncta is not obvious in the images. Perhaps it is a color-scale issue.

The line profiles in Fig 7f are more convincing; however, that is a single puncta and does not inherently guarantee that all/most puncta have that phenotype.

Membrane Tethering: Fig 7b/c. To clarify: was the tether data taken during phagocytosis taken from anywhere on the cell or from near the vicinity of the phagosome?

Membrane lifting around Actin/FcR adhesions: Why does the membrane lift? How does this work mechanically? Is the actin like in a podosome, pushing outward? What is the imagined mechanism by which myosin 1e/1f are involved?

Text:

p. 5 Unlike the rest of the manuscript, which read beautifully, I paused at this somewhat unclear and awkward sentence in the introduction. But this may be a matter of taste and I am fine if it remains the same: "Whether actin polymerization is regulated biophysically to successfully close the phagocytic cup has yet to be explored."

Reviewer #2:

Remarks to the Author:

I have reviewed the paper "Membrane-Cytoskeleton Mechanical Feedback Mediated by Myosin-I Controls Phagocytic Efficiency" by Barger et al. In this work, the authors provide exciting mechanistic insight into the role myosin-I in the formation of phagosomes by macrophages. While a number of myosin studies have been conducted in phagocytosis, this reviewer is unaware of previous work examining the role of myosin-I in phagocytosis. While myosin-I does not appear to be required for the uptake of plastic beads, it does appear to contribute to the efficiency of phagocytosis and as the authors demonstrate, MYO-1f and MYO-1e both contribute through activities localized to the tips of the forming phagosome. Overall, this reviewer is enthusiastic about the manuscript, but the manuscript should be significantly improved by careful interpretation of the data/experimental models and clarity of presentation prior to publication. Major concerns:

pp. 7

"With myo1e and myo1f binding both actin filaments and membrane lipids, the lack of these myosins could potentially affect FcR clustering and activity at the plasma membrane or the ability of cells to assemble actin to form the phagocytic cup. To test whether any of these mechanisms could account for the observed phagocytosis defect, we first assessed downstream signaling from FcRs by stimulating BMDM with anti-FcR antibody followed by antibody cross-linking at 37°C for different periods of time. "

This experiment a remarkably poor model choice for FcR signaling pertaining to phagocytosis and to this reviewer provides little evidence that signaling for phagocytosis is unaltered. Cross-linked FcRs are possibly a reasonable model for signaling from soluble immune complexes but without a surface, the comparison with phagocytosis is nearly irrelevant. It would be okay to conclude that Syk activation is still possible in the Myo1e and Myo1f dKO, and that seems sufficient for the study, however it seems brazen and premature to conclude that signaling for phagocytosis is unperturbed. The work of Swanson and colleagues has demonstrated multiple stages of signaling that would likely be affected by the Myo1 defects (Zhang et al PNAS 2010). Second, beads are far from a physiological target – and they are internalized absent antibody – so concluding that FcR clustering, binding and activation is a very soft statement for hard plastic beads. The use of sheep red blood cells would be a compelling alternative – and may lead to a different result altogether.

Additionally, many of the experiments rely on a 'frustrated phagocytosis' model, where glass has been coated non-specially with IgG. This is an old model, but it has its limitations. In particular, macrophage spread fine on glass, and generally with kinetics and morphology that is indistinguishable between bare glass and IgG coated glass. I don't feel that this precludes the conclusions of the role of Myo1e&f, but caution is urged in terms of assuming adhesion is mediated predominantly by FcRs.

Figure 6 g-i seem to be floating in space. It is entirely unclear to this reviewer what the point of these panels are. From the results it seems that they are intended to capture the wave dynamics, however, without a comparison to wild-type macrophages spreading on glass, there is nothing to compare this with and the conclusions from these results are completely lost.

Lastly, and perhaps most importantly, the authors propose a model for how Myo1e&f could be regulating FcR microclusters and actin polymerization during cup formation and closure. The concept of 'lifting membrane' near the microclusters is very intriguing, although the idea comes very late in the paper and feels underdeveloped. A pictorial model accompanied by a clear description of how the results support the model would be very helpful.

"By lifting membrane around individual adhesions, myo1e/f prevents adhesion fusion and expansion, which results in slower internalization."

Confusing sentence. But also an important one. Please expand this idea to make it clear. Lastly, the manuscript readability is overall good. Consideration of paragraph topic sentences, would strengthen the paper. Often the topic sentences are ill-formed or appended to the previous paragraph and therefore derail the reader.

Minor Concerns:

pp. 3

"Whether actin polymerization is regulated biophysically to successfully close the phagocytic cup has yet to be explored."

This statement is enigmatic, especially given the topic sentence of the paragraph and the extensive literature around actin polymerization in phagocytosis and the dozens of papers that exist on biophysical mechanisms including force generation....?

"Overall, this work provides an unexpected biophysical explanation for how actin dynamics are precisely controlled to promote extension and closure of the phagocytic cup."

pp 12 "This is the first report of myosin 1s as components of FcR adhesions."

A topic sentence that relates to the content of the paragraph that follows would be much more helpful to the reader.

"Another interesting suggestion for myosin II function is the exclusion of specific cell receptors, such as CD47 or phosphatase CD45, from the phagocytic cup to allow proper downstream signaling 71, 72. "

CD47 is the target-side ligand for the macrophage receptor SIRPα and CD45 as a phosphatase that is thought to be excluded by size, not a ligand interaction. Please revise.

The frustrated phagocytosis experiments in Fig 3 are confusing. How would the Fc receptors move on a rigid surface with attached IgG.

Fig 6 e and f. Clearly state if the results in e are statistically significantly different or not. If not, leaving off the test is not a valid way to represent the result.

"We next questioned whether myo1e/f might be regulating membrane tension locally at FcR adhesion sites, and, at the same time, restricting adhesion expansion and lateral sliding." Should be topic sentence for paragraph on bottom of pp 19.

Reviewer #3:

Remarks to the Author:

In their manuscript entitled "Membrane-cytoskeleton mechanical feedback mediated by Myosin-I controls phagocytic efficiency," Barger et al. present intriguing findings indicating that myosins-1 play a role in organizing actin at sites where phagocytic cells are adhering to targets during engulfment. The manuscript is supported by a substantial amount of experimental data utilizing many different and complementary techniques. Demonstrating that Myosin-1e functions mainly as an actin organizer during this process represents a considerable conceptual advance.

While the findings seem overall well supported and interesting, this manuscript could be improved significantly by reorganization. The paper, as it is currently written, is a bit fatiguing to read. The figure panels, particularly in the supplemental figures, are not always referred to in order, so reading the manuscript became an exercise in paper shuffling. For example, all panels of supplementary figure 2 are referred to before the first mention of supplementary figure 1F. The authors also guide the reader through many pages of negative data before presenting their more

interesting positive data. Reorganization of figure panels so that they are referred to in order and a more concise discussion of negative data would go a long way towards streamlining the manuscript and bringing forth the important findings.

I also have a number of experimental suggestions to improve the study. They are listed below.

1) The results in figure 1h suggest that phagocytosis occurs more slowly in dKO cells. It would be interesting and informative to see live imaging of the dynamics of phagocytic cup progression and closure and quantification of the timing.

2) The results in supplementary figure 1f are difficult to interpret because the experimental procedure is not well described. Why were the cells stimulated with anti FcR antibody when for most other experiments (e.g. bead engulfment) IgG was used? What is antibody cross-linking? This procedure should be explained more fully in the methods.

3) I do not agree with the interpretation of the experiments presented in supplementary figure 2. First, the authors claim that these data indicate that Myo1e and Myo1f may bind to PI(3,4,5)P3, but the probe that they are using for visualization, AKT-PH, also binds to PI(3,4)P2 by the authors own admission. The experiments presented do not exclude a role for PI(3,4)P2. The results shown using the PI3K inhibitor are incomplete. Because the authors do not show any evidence that they have been successful in reducing PI(3,4,5)P3 levels, a reasonable interpretation of the mild Myo1e localization defect is simply that drug treatment was ineffective.

4) The authors analyze the colocalization of Myo1e/f in relationship to FcγRIIA during frustrated phagocytosis, but it would be informative to see a two-color imaging experiment done on cells undergoing actual phagocytosis, as is done in figure 3 f and g. To give the reader a better idea of how myo1e/f are arranged in relation to actin and FcR during phagocytosis, it would help to see some kind of quantification of the types of micrographs shown in figure 3 f and g. This would demonstrate how regular the positioning of Myo1 is during this process.

5) On page 16, the authors say that Myo1e/f are negative regulators of the arp2/3 complex. This type of claim really requires biochemical evidence for support, and I doubt that Myo1e/f are directly opposing arp2/3 activation. It may be more precise to say that Myo1e/f corral or constrain arp2/3-mediated actin assembly.

6) The experiments presented in figures 4 and 5 seem incomplete because they look almost exclusively at actin. Are there FcR cluster aggregates that underlie the actin aggregates?

7) The SIM data in the same figure are also difficult to interpret. The wildtype cell shown doesn't seem like a good representative image, because it doesn't look like it has a true actin wave as the wildtype cells in panel A do. Maybe the clusters in this cell would look more like a wave if the cell was imaged with conventional microscopy? If that is the case, a picture of the same cell imaged with conventional, diffraction limited microscopy should be provided.

8) Statistical information is missing for figure 6h. How many cells were measured to generate this plot, and how many measurements were taken per cell? I'd prefer to see a more conventional scatter plot so that each data point is actually visible.

9) In figure 7, how big are the areas of elevated membrane that surround the actin spots? Are the actin spots always right in the middle of the elevated membrane? I would like to see more quantification of these data. At the minimum, I'd like to see a version of figure 7f where the same measurement was done on multiple cells and the results normalized and averaged. This would strengthen the case that the membrane-lifting phenomenon is actually absent in the dKO cells.

Non-experimental suggestions

1) The title of this work could be more descriptive. The manuscript is mostly concerned with organization of F-actin at FcR adhesion sites, but that isn't mentioned in the title. I don't know what "membrane-cytoskeleton mechanical feedback" means. Live imaging of phagocytosis in the dKO cells (see above) may allow the authors to say something more articulate than simply saying myosin 1 "controls phagocytic efficiency."

2) A minor point: it seems strange to me that Nambiar, McConnell, and Tyska (PNAS, 2009) is not cited here, because it seems very relevant to the work.

3) Figure 1i is not cited in the main text and it is hard to see phagocytosed beads since the image

is so small. The non-phagocytosed beads should be highlighted as well. For visualization purpose it might be better to move it to the supplementary Figures and enlarge the image.

4) In figure 1 and supplementary figure 1 it was confusing to see part of the results of Myo1e and Myo1f mixed in both figures for the reader. I would suggest only showing the results of Myo1e or Myo1f in the main Figure and move one of the results of Myo1e or Myo1f in the supplement.

5) On page 7 there is an error in referencing figures. The text refers to "Figure 1h-l", there is no figure 1l.

6) It would be helpful to summarize the findings in a model figure at the end, as was partially done in Figure 3h. That diagram is very helpful!

Reviewer #4:

Remarks to the Author:

In this work by Barger et al. The role of the myosin motors myosin 1e and myosin 1f are investigated in phagocytosis of antibody-coated particles in macrophages. The authors find that these motors which link the actin cytoskeleton to the plasma membrane are concentrated near F-actin sites that form around the phagocytic cups and FcR-actin adhesions. Double knock-outs for myosin1e/f showed slower and less uptake of 6 micron Ig-coated beads. These cells had lower membrane tensions and malformed large actin adhesions. Thus, the authors propose that the inability to form the correct actin networks in these myosin mutants could be decreasing the proper assembly of actin near FcR-based adhesions necessary for the completion of phagocytosis. In general this paper is well done, comprehensive, and interesting. The figures are quite nice and the imaging/experiments are impressive. I have several questions and comments regarding this work that might be addressed to improve the manuscript.

1. In figure 1h the major (most dramatic) effect of the dKO appears at the 15 min time point and less so at the 60 min time point with only a ~30% reduction in cells with particles compared to wild-type. This effect isn't dramatic. If myosin1e/f is central for phagocytosis why the relatively minor reduction in phagocytic activity? Are there other situations where the myosin would be more critical? Clearly, the protein is involved in fine-tuning and enhancing phagocytic uptake but could the authors comment on the relative importance of these motors in the overall process?

2. How is this myosin targeted to the FcR clusters? The authors note that a TH2 domain mutation is key for localization. Does this domain interact with any specific proteins? Also, does re-expression of this mutant fail to rescue the dKO phenotype in Figure 1h? Does expression of just the TH2 domain block phagocytosis?

3. The "lifting" observation in Figure 7 is not clearly describe or explained in the text or figures. Furthermore, I cannot see how this "lifted" TIRF membrane spatially relates to the other fluorescent signals (myo or actin) shown in figure 7. Additional quantitative plotting or analysis above a single line scan that is currently shown would help this figure. As this concept is a major feature of the author's model, and as it stands now is merely observational, it would be nice to quantitatively analyze this effect more extensively.

As this was observed in the Orsi 1983 paper it would also be nice to incorporate those older models more clearly and prominently in this paper.

4. The paper would benefit from a general cartoon figure where the authors clearly propose their model of adhesion and cup formation with the "lifting" idea and the role of these myosins in that process.

5. Why is the initial phagocytic cup not impacted by myosin dKO but the later stages (closure?) of phagocytosis impacted? If the adhesion/tension and lifting behavior is key I would expect a more

pronounced change throughout phagocytosis. Some discussion or model for this part of the process would benefit the paper and help with future experimental tests of this model.

We thank the reviewers for their insightful comments and constructive criticisms and we are glad that all reviewers found this manuscript to be of interest and describing novel findings. Following reviewers' suggestions, we have added further experimental data and performed additional analyses; we have also revised the text and figures to clarify our observations and hypothesis and to streamline the figure/text organization. All reviewers suggested that we should provide more details regarding the proposed mechanism for myosin-I functions and include a summary graphical representation to depict our model. We have therefore changed the discussion substantially and added a schematic representation of the model (Fig. 7i). We hope that all of these changes have addressed the concerns raised by the reviewers and improved the clarity of the manuscript. The key changes to the text have been highlighted in the redlined version of the manuscript file.

Below, we address the individual reviewers' comments:

Reviewer #1 (Remarks to the Author):

The manuscript by Barger et al entitled 'Membrane-cytoskeleton mechanical feedback mediated by Myosin-I controls phagocytic efficiency' is a well written and interesting report that addresses for the first time the role of myosins 1e and 1f in the phagocytic cup. The manuscript is rich with convincing data and the overall result should be relevant to wide range of researchers. The authors employed an impressive range of sophisticated techniques (FRAP, AFM, optical tweezers, traction force, FACS, SIM, correlative fluorescence and EM, etc.) to perform elegant experiments in order provide a detailed characterization of myosin 1e/1f. They used a combination of standard phagocytosis and frustrated phagocytosis assays with primary macrophages to achieve their goals. The work presents compelling data to support a series of claims which together culminate to support the authors' hypothesis that the work in total identifies a novel role for class 1 myosins in coordinated adhesion turnover during phagocytosis. Their main results include what is effectively a summary of the titles listed throughout the manuscript. I will briefly summarize the most important findings:

1. Myo 1e/f are required for efficient phagocytosis and are co-localized with at the front of the phagocytic cup with actin (actually slightly in front of the actin)
2. Intriguingly myo1e/1f are not required for FcR signaling, target binding or phagocytic cup formation. They are not involved in contraction during phagocytic cup closure nor do they participate in focal exocytosis.
3. Myo1e/1f localize to FcR-actin adhesions during frustrated phagocytosis and that their presence helps prevent the overgrowth of actin potentially by lifting the cell plasma membrane from the surface. Indeed if cells lack myo1e/1f, this results in the formation of disorganized actin-based adhesions and the formation of denser phagocytic cups. Overall, the work is a very thorough study which is well assembled and reported. There are few instances were clarifications or corrections would improve the document – described below. Further I have one main concern regarding their second claim (in the abstract): The manuscript's second main claim is that the results 'support a model for a membrane-tension based feedback mechanism for

phagocytic cup closure' is less clear. This is my largest critique of the manuscript – that this claim would be better supported by an expanded discussion of their proposed mechanism for how Myo1 regulates membrane tension. Specifically, it would be helpful if they layout a step-by-step timeline of events, being sure to include when and how myo1 regulates tension and why that cascades to down-stream process to complete phagosome closure. A schematic would help tremendously. Unlike the rest of the manuscript which reads very smoothly and clearly with good messaging, this other key claim is less well fleshed out. The mere fact that the summary of the key data results above do not directly include tension shows why / how this point might be less clear or not well supported.

In response to Reviewer 1, we have changed Figure 7 detailing the role of myo1e/f in regulating membrane tension during phagocytosis. We have added more details regarding our observations of myo1e/f at areas of membrane lifting (pg. 20, Fig 7e-i, Supplementary Movie 15 and 16), new quantifications (Fig. 7h) and included a schematic drawing of our proposed model (Fig 7i). We have also modified the discussion to more fully explain our hypothesized mechanism.

Other minor feedback:

In some cases the authors give the number of cells assessed (N) and in other cases they don't. For all data presented, the authors should report the number of cells measured in all reported results, so that readers have an indication of what the actual spread in values are and not just the uncertainty of the mean. For example, Figure 1h, j, k do not include the number of cells. Similar, when quantifying actin adhesions (area, height), the total number measured should be reported (see Fig 4d-f)

We agree with the reviewer that this is important information. We have therefore modified our Figures and Figure Legends to include Ns in all experiments presented.

Now some feedback organized around the figures:

Figure 1

- Figure 1b shows a cross section for Fig 1a and says it corresponds to dotted line in Fig 1a. I don't see that line – at least not in my print out

This line has been thickened.

- Fig 1d has reversed colors for myosin and actin relative to many of the other images in the same figure, including 1a-c and 1f; I recommend switching them for consistency and ease of digestion

Thank you for your suggestion, and we completely understand your point. However, we would like to keep the colors consistent with the type of fluorescent protein used in a particular experiment (for example, we try to always use green color to designate EGFP-tagged proteins often paired with magenta as a color blind friendly option).

- Fig 1f – I don't find this figure very informative. It isn't clear that the yellow arrow is pointing to the edge of the phagocytic cup – the entire bead appears to be coated with fluorescence.

This image depicts a cup that is facing upward, so it is impossible to see the leading edge given the orientation of the cup. It is not a closed cup/phagosome, as indicated by the amount of actin labeling surrounding the circumference of the bead. Endogenous myo1e is also outlining the bead, which is what we are trying to point out in this figure. We have changed the description in the Figure Legend (p. 30) to make this clearer.

- Fig 1h – as stated above, it would be beneficial to have data on the total number of cells measured rather than just the % phagocytic cells. I won't repeat this for every figure.

The N has been added in the Figure Legend as well as for all the other missing Ns (as suggested above), however for this specific case we believe the percentage of phagocytic cells is easier to report than the large number of cells we quantified over the 10-18 FOV imaged at 20X magnification.

- Fig 1i – say in the caption what cell type these are BMDM or RAW264.7

This information (BMDM) has been added in the Figure Legend (p.30).

Figure 4.

- What is the criteria to distinguish between a diffuse actin wave and aggregated wave? How was the data in Fig 4b evaluated? In Fig 4c, why are the puncta distributed all over the cell rather than in a diffuse ring/wave? Is this expected?

To expand on the criteria used to distinguish between diffuse and aggregated actin waves, we have added a judging guide/set of examples to the Supplementary Figure 7d. This table displays examples of what the authors considered a diffuse or aggregated actin wave, along with a brief description of each. All cells analyzed were fixed and stained using the same procedure on the same day. Before being judged, microscopy files were shuffled and renamed using the ImageJ Blind_Analysis macro, so that evaluators were blinded to the genotypes of cells analyzed. For Figure 4c, we have chosen different (and 2 instead of 1) SIM images to better illustrate the actin wave in WT/dKO cells. These are more typical representations than the images in the first submission.

- Why isn't an image shown from the 3DSIM to show the actin adhesion height? Fig 4e? It would be useful to show so the reader has a sense of the quality of the data.

As suggested, we have added an xz image of the WT and dKO actin waves below in Figure 4c that shows the difference in phagocytic adhesion height. The Supplementary Movie of the EM images of the actin wave (Supplementary Movie 12) also provides readers with some idea of the height of these structures.

- What kind of resolution does the 3DSim afford in z? It is worth stating this since it is a relatively new technique and it could vary from setup to setup.

3D SIM improves resolution by a factor of 2 in the 3 dimensions – bringing resolution limits close to 100nm in x/y and around 300nm in z. This information has been added to the manuscript (pg. 14).

Figure 6

Fig6c: is it SEM or St dev shown in the error bars?

The error bars in Figure 6c represent standard deviation. This has been clarified in the Figure Legends (p.33).

It's not clear what is being measured and reported in the plots in Fig 6 h and i. How is each point in Fig 6h determined? Is it some pixel bin located at the cell periphery? Is it an average for the whole cell? Similarly, how was the data averaged for Fig 6i is that one cell or an average of many?

To avoid confusion, we are now including only the graph of averages (former Figure 6i) as Figure 6h and not presenting the cloud graph of all the data (former Figure 6h). This figure was generated from a single dKO cell undergoing many cycles of wave extension and retraction. We've added the movie of this measured cell as Supplementary Movie 14 to aid in our presentation of this figure. We chose to analyze this single dKO cell because it displayed dynamic behavior in terms of wave protrusion or retraction. Because cells exhibit variability in dynamics when performing frustrated phagocytosis and variability in EGFP-actin expression given our use of transient transfection, using a large number of cells for this analysis does not necessarily add power. Nevertheless, we conducted the same analysis on more dKO cells (n=7) with similar results. This graph is presented as Supplementary Figure 10, however the results are normalized in order to give similar statistical weight for each cell, regardless of EGFP-actin expression or dynamics.

To clarify, our main goal for these figures was to test whether increased actin intensity in the wave was inversely correlated with wave speed in dKO cells. To extract these data, first the exterior of the actin wave was traced by hand throughout the time-lapse. From these points, wave boundary speed is determined radially by computing the distance change between two consecutive frames. Actin wave protrusion was defined as a positive number, while wave retraction was negative. The corresponding mean actin intensity for the same angles is averaged from the wave edge boundary to 30 pixels (~4um) inward, which encompasses the whole actin wave (Supplementary Movie 14 shows both lines used). The cloud graph in the previous manuscript showed all data, from each angle (between 0 and 359 degrees) over the whole time-lapse. The averaged data (current Figure 6h) with corresponding standard deviation error bars is obtained from individual angle measurements throughout the time-lapse. We have modified the explanation of this figure in the Results section (pg. 17-18) and added a detailed description of the steps for analysis in the Methods section (p.47-48).

Figure 7

I found Fig 7 hard to interpret. In particular the "donut" of stained membrane around the actin puncta is not obvious in the images. Perhaps it is a color-scale issue.

The line profiles in Fig 7f are more convincing; however, that is a single puncta and does not inherently guarantee that all/most puncta have that phenotype. Membrane Tethering: Fig 7b/c. To clarify: was the tether data taken during phagocytosis taken from anywhere on the cell or from near the vicinity of the phagosome? Membrane lifting around Actin/FcR adhesions: Why does the membrane lift? How does this work mechanically? Is the actin like in a podosome, pushing outward? What is the imagined mechanism by which myosin 1e/1f are involved?

These are all great questions, and we hope we have addressed them to Reviewer's satisfaction in the Results section (pg. 20), updated Figure 7e-i, and the Discussion. We have updated Figure 7 images to more clearly illustrate the membrane lifting using a different color scheme. In addition, we have included movies of the observed behavior in WT and dKO cells (see Supplementary Movie 15 and 16). The previous line-scan figure was actually an average of line scans of >80 adhesions in multiple cells, rather than a single line scan. Based on the reviewer's comments, we decided to replace this line scan with a systematic analysis of averaged actin and membrane fluorescence intensity in the area surrounding each adhesion (rather than using a single line scan through each adhesion) (Fig. 7h). This analysis combines data from many adhesions of multiple cells, as indicated in the Figure legend. We measured membrane tension using the tether pulling assay specifically near the forming phagocytic cup. We have highlighted this in the text and changed the cartoon schematic (Fig. 7c) to better represent this set-up. We have included a schematic of our proposed mechanism in Figure 7i and further explored the questions raised by Reviewer 1 in the amended Discussion section.

Text:

p. 5 Unlike the rest of the manuscript, which read beautifully, I paused at this somewhat unclear and awkward sentence in the introduction. But this may be a matter of taste and I am fine if it remains the same: "Whether actin polymerization is regulated biophysically to successfully close the phagocytic cup has yet to be explored."

We have changed this sentence (p.3 in the updated text). Thank you for your feedback.

Reviewer #2 (Remarks to the Author):

I have reviewed the paper “Membrane-Cytoskeleton Mechanical Feedback Mediated by Myosin-I Controls Phagocytic Efficiency” by Barger et al. In this work, the authors provide exciting mechanistic insight into the role myosin-I in the formation of phagosomes by macrophages. While a number of myosin studies have been conducted in phagocytosis, this reviewer is unaware of previous work examining the role of myosin-I in phagocytosis. While myosin-I does not appear to be required for the uptake of plastic beads, it does appear to contribute to the efficiency of phagocytosis and as the authors demonstrate, MYO-1f and MYO-1e both contribute through activities localized to the tips of the forming phagosome. Overall, this reviewer is enthusiastic about the manuscript, but the manuscript should be significantly improved by careful interpretation of the data/experimental models and clarity of presentation prior to publication.

We thank the Reviewer for this positive evaluation of our work.

Major concerns:

pp. 7

“With myo1e and myo1f binding both actin filaments and membrane lipids, the lack of these myosins could potentially affect FcR clustering and activity at the plasma membrane or the ability of cells to assemble actin to form the phagocytic cup. To test whether any of these mechanisms could account for the observed phagocytosis defect, we first assessed downstream signaling from FcRs by stimulating BMDM with anti-FcR antibody followed by antibody cross-linking at 37°C for different periods of time. “

This experiment a remarkably poor model choice for FcR signaling pertaining to phagocytosis and to this reviewer provides little evidence that signaling for phagocytosis is unaltered. Cross-linked FcRs are possibly a reasonable model for signaling from soluble immune complexes but without a surface, the comparison with phagocytosis is nearly irrelevant. It would be okay to conclude that Syk activation is still possible in the Myo1e and Myo1f dKO, and that seems sufficient for the study, however it seems brazen and premature to conclude that signaling for phagocytosis is unperturbed. The work of Swanson and colleagues has demonstrated multiple stages of signaling that would likely be affected by the Myo1 defects (Zhang et al PNAS 2010). Second, beads are far from a physiological target – and they are internalized absent antibody – so concluding that FcR clustering, binding and activation is a very soft statement for hard plastic beads. The use of sheep red blood cells would be a compelling alternative – and may lead to a different result altogether.

Thank you for your feedback on this experiment. In response, we have reworded our conclusions regarding our findings because it would indeed be premature to state that all phagocytic signaling remains unaffected in our knock out cells. Although we agree with Reviewer 2 that sRBCs would be a more physiologically relevant target than polystyrene beads, we are hesitant to change our targets now given the rest of our experimental data. We initially chose polystyrene beads due to their uniform structure and multiple sizes available. We specifically used this test of FcR stimulation because we wanted to stimulate cells in the absence of a target, which would cause a change in

cell shape and a rearrangement of the actin cytoskeleton and complicate the analysis. While we agree with the reviewer that this experiment is limited in its ability to faithfully recapitulate all phagocytic signaling, it has been used in multiple papers to assess activation of signaling pathways downstream of FcR, specifically in relation to phagocytosis (e.g. Fitzer-Attas et al. (2000) *JEM*; Cao et al (2004) *JJ*; Gu et al. (2003) *JCB*) We believe this method differs from testing the signaling of soluble immune complexes in that the added antibodies in such a test are usually heated to form aggregates (e.g. Dale et al. (2009) *JJ*; Sobata et al. (2005) *JJ*; Mao et al (2009) *JCB*). Thus, we have modified our description of this figure in the text (p.7), but still feel it adds value to our manuscript as we can show that the aggregated actin polymerization in the dKO macrophages is occurring downstream of specific FcR kinase activity and as we hypothesize is a result of altered membrane biophysical regulation.

Additionally, many of the experiments rely on a ‘frustrated phagocytosis’ model, where glass has been coated non-specially with IgG. This is an old model, but it has its limitations. In particular, macrophage spread fine on glass, and generally with kinetics and morphology that is indistinguishable between bare glass and IgG coated glass. I don’t feel that this precludes the conclusions of the role of Myo1e&f, but caution is urged in terms of assuming adhesion is mediated predominantly by FcRs.

We agree with Reviewer 2 about the limitations of the ‘frustrated phagocytosis’ model and have therefore tried to replicate our findings in 3D whenever possible (e.g. Figure 3f-g, Figure 4f-h, Supplementary Figure 5c-f). While macrophages do indeed spread on glass, fibronectin and IgG coating we have quantified the percentage of cells that form actin waves on these substrates and included this information in the manuscript in Supplementary Figure 7a-b. We show that these structures chiefly form in response to IgG and rarely on other surfaces, reinforcing this system as an appropriate model for studying actin assembled by phagocytic signaling.

Figure 6 g-i seem to be floating in space. It is entirely unclear to this reviewer what the point of these panels are. From the results it seems that they are intended to capture the wave dynamics, however, without a comparison to wild-type macrophages spreading on glass, there is nothing to compare this with and the conclusions from these results are completely lost.

We are now presenting only the averaged graph (former Fig 6i) from this analysis and not the cloud graph (former Fig. 6h). Indeed, the purpose of this analysis was to understand the actin wave dynamics in the dKO macrophages, namely, is increased actin intensity in the wave inversely correlated to actin wave speed? We are interested in actin wave speed as a proxy for the kinetics of phagocytic cup extension during phagocytosis. The majority of the data measured from the dKO cell shows that a stalled actin wave corresponds with increased actin wave intensity. Because the WT macrophages do not exhibit the aggregated actin wave phenotype and instead produce actin adhesions of limited variability (see Figure 4d and 4e), analyzing their dynamics would not be useful to answer this question (or necessary for a side-by-side comparison). In the text, we have tried to explain our reasoning for this analysis more clearly. We have also further clarified how this analysis was performed (p. 17-18 of the Results and the corresponding Methods section) and included a supplementary

video of the measured dKO cell (Supplementary Movie 14) and a normalized quantification of 7 dKO cells (Supplementary Figure 10).

Lastly, and perhaps most importantly, the authors propose a model for how Myo1e&f could be regulating FcR microclusters and actin polymerization during cup formation and closure. The concept of ‘lifting membrane’ near the microclusters is very intriguing, although the idea comes very late in the paper and feels underdeveloped. A pictorial model accompanied by a clear description of how the results support the model would be very helpful.

“By lifting membrane around individual adhesions, myo1e/f prevents adhesion fusion and expansion, which results in slower internalization.”
Confusing sentence. But also an important one. Please expand this idea to make it clear.

We have attempted to expand on this idea in the Figure 7 Results section and the Discussion. We have also added more descriptions (pg. 19-20) and quantification of our membrane lifting observations (Fig. 7h) in RAW and primary cells in the form of new figures and supplementary movies (Supplementary Movies 15 and 16). A schematic detailing our hypothesized mechanism has also been added (Fig. 7i).

Lastly, the manuscript readability is overall good. Consideration of paragraph topic sentences, would strengthen the paper. Often the topic sentences are ill-formed or appended to the previous paragraph and therefore derail the reader.

Thank you for bringing this to our attention. We have attempted to re-write many paragraph opening sentences (new or relocated topic sentences are highlighted in red).

Minor Concerns:

pp. 3

“Whether actin polymerization is regulated biophysically to successfully close the phagocytic cup has yet to be explored.”

This statement is enigmatic, especially given the topic sentence of the paragraph and the extensive literature around actin polymerization in phagocytosis and the dozens of papers that exist on biophysical mechanisms including force generation....?

Agreed. This introductory paragraph has been changed.

“Overall, this work provides an unexpected biophysical explanation for how actin dynamics are precisely controlled to promote extension and closure of the phagocytic cup.”

pp 12 “This is the first report of myosin 1s as components of FcR adhesions.”

A topic sentence that relates to the content of the paragraph that follows would be much more helpful to the reader.

Agreed. In this second submission, we have attempted to make all topic sentences relate to the content of the paragraph. Thank you for this suggestion.

“Another interesting suggestion for myosin II function is the exclusion of specific cell receptors, such as CD47 or phosphatase CD45, from the phagocytic cup to allow proper downstream signaling^{71, 72}. “

CD47 is the target-side ligand for the macrophage receptor SIRPα and CD45 as a phosphatase that is thought to be excluded by size, not a ligand interaction. Please revise.

This paragraph has been removed to make room for other discussion, but the authors appreciate your correction.

The frustrated phagocytosis experiments in Fig 3 are confusing. How would the Fc receptors move on a rigid surface with attached IgG.

Yes, as the Reviewer correctly noted, due to the attachment of IgGs to the glass substrate, the FcR clusters on the cell do not move and are fixed on the substrate. This is the opposite of what is seen during frustrated phagocytosis experiments using supported lipid bilayers, where FcR clusters are able to move in the plane of the membrane. Based on our observations, when the actin-myosin wave passes and colocalizes with the FcR clusters, the fluorescence intensity of individual FcR puncta increases. As the wave moves on, the fluorescence intensity of these puncta dims, but they do not disassemble completely (unlike the actin or myosin puncta). These clusters clearly form upon contact with the coated coverslip and are not pre-assembled on the macrophage. We have provided an additional movie depicting this process (Supplementary Movie 7).

Fig 6 e and f. Clearly state if the results in e are statistically significantly different or not. If not, leaving off the test is not a valid way to represent the result.

The statistics for this Figure have been added.

“We next questioned whether myosin might be regulating membrane tension locally at FcR adhesion sites, and, at the same time, restricting adhesion expansion and lateral sliding.”

Should be topic sentence for paragraph on bottom of pp 19.

Thank you for your suggestion – we have changed this in the manuscript.

Reviewer #3 (Remarks to the Author):

In their manuscript entitled “Membrane-cytoskeleton mechanical feedback mediated by Myosin-I controls phagocytic efficiency,” Barger et al. present intriguing findings indicating that myosins-1 play a role in organizing actin at sites where phagocytic cells are adhering to targets during engulfment. The manuscript is supported by a substantial amount of experimental data utilizing many different and complementary techniques. Demonstrating that Myosin-1e functions mainly as an actin organizer during this process represents a considerable conceptual advance.

We thank the reviewer for these encouraging comments.

While the findings seem overall well supported and interesting, this manuscript could be improved significantly by reorganization. The paper, as it is currently written, is a bit fatiguing to read. The figure panels, particularly in the supplemental figures, are not always referred to in order, so reading the manuscript became an exercise in paper shuffling. For example, all panels of supplementary figure 2 are referred to before the first mention of supplementary figure 1F. The authors also guide the reader through many pages of negative data before presenting their more interesting positive data. Reorganization of figure panels so that they are referred to in order and a more concise discussion of negative data would go a long way towards streamlining the manuscript and bringing forth the important findings.

In response to Reviewer 3, we made sure that all Supplementary Figures appear in the order in which they are mentioned.

Reviewer 3 certainly makes a good point, as we realize we are putting most of the exciting data at the end of the manuscript. However, we also felt that since there were several logical possibilities for how myosin-I could impact phagocytosis, we had to consider them and test them one by one, and that we needed to fully report on the outcomes of our investigation of the potential avenues for myosin-I function. While appreciating this critique, unfortunately, we could not conceive of a better organization for the manuscript without omitting important information, including additional experiments and clarifications proposed by other reviewers.

I also have a number of experimental suggestions to improve the study. They are listed below.

We thank Reviewer 3 for the suggestions and did our best to address some specific points (see below):

1) The results in figure 1h suggest that phagocytosis occurs more slowly in dKO cells. It would be interesting and informative to see live imaging of the dynamics of phagocytic cup progression and closure and quantification of the timing.

We agree with Reviewer 3 on this point. However, imaging phagocytosis is challenging due to the 3D nature of the process (which often can't be captured in one optical section) and photosensitivity of the cells (making collecting z-stacks

phototoxic). The low transfection rate of primary macrophages makes the experiments even more difficult. Based on the findings from our bulk assay (Figure 1h), we believe that while cup closure in the dKO is slower, it is not dramatic, and, therefore, we would need to capture a very large number of phagocytic events to show a statistically significant difference in progression/closure of individual phagocytic cups, similar to that observed in the bulk assay.

2) The results in supplementary figure 1f are difficult to interpret because the experimental procedure is not well described. Why were the cells stimulated with anti FcR antibody when for most other experiments (e.g. bead engulfment) IgG was used? What is antibody cross-linking? This procedure should be explained more fully in the methods.

We have attempted to further clarify this experimental procedure in both the Results and Methods sections. Antibody crosslinking (in this experiment) is when antibodies that are bound by Fc receptors on the cell surface are exposed to and bound by a secondary antibody via the complement region of the first antibody. In immune cells, this causes FcR activation and has been often used to assay initial Fc receptor signaling in the absence of a target (see Fitzer-Attas et al. (2000) *JEM*; Cao et al (2004) *Jl*; Gu et al. (2003) *JCB* etc). We chose to test this initial FcR response of WT and dKO macrophage along with bead binding to rule out the role of myo1e/f in the early stages of phagocytosis.

3) I do not agree with the interpretation of the experiments presented in supplementary figure 2. First, the authors claim that these data indicate that Myo1e and Myo1f may bind to PI(3,4,5)P3, but the probe that they are using for visualization, AKT-PH, also binds to PI(3,4)P2 by the authors own admission. The experiments presented do not exclude a role for PI(3,4)P2.

Reviewer 3 makes a good point. We have attempted to do the same experiment with the published lipid sensor, TAPP1, which solely recognizes PI(3,4)P2 (Kimber et al. (2002)). We have included a description of the data in the text (p. 5-6) and added the incidence of colocalization with this sensor in the Supplementary Figure 2b.

The results shown using the PI3K inhibitor are incomplete. Because the authors do not show any evidence that they have been successful in reducing PI(3,4,5)P3 levels, a reasonable interpretation of the mild Myo1e localization defect is simply that drug treatment was ineffective.

To test the possibility suggested by this Reviewer, we have repeated these experiments including cells expressing EGFP-AKT-PH as a control. When LY249002 is added to the cells expressing the AKT-PH, the cup/cytosol fluorescence intensity drops significantly, suggesting that the inhibitor is effective (Supplementary Figure 2d).

4) The authors analyze the colocalization of Myo1e/f in relationship to FcγRIIA during frustrated phagocytosis, but it would be informative to see a two-color imaging experiment done on cells undergoing actual phagocytosis, as is done in figure 3 f and g.

To give the reader a better idea of how myo1e/f are arranged in relation to actin and FcR during phagocytosis, it would help to see some kind of quantification of the types of micrographs shown in figure 3 f and g. This would demonstrate how regular the positioning of Myo1 is during this process.

Using lattice light sheet microscopy, we have attempted to perform live-cell imaging on RAW macrophages co-transfected with FcR and myo1 or myo1 and Lifeact. Unfortunately, we were unable to capture well-transfected (with sufficient signal-to-noise for myo1 or FcR) cells engulfing beads. We did manage to capture cells nicely expressing Lifeact engulfing beads and forming adhesions, similar to those shown in the fixed images of cups in Figure 3. This data has been added as Figure 3i and Supplementary Movie 11. As shown in this video, the adhesions appear most clearly at certain times during phagocytic internalization and during cup closure seem to coalesce. Because of this, it is slightly difficult to quantify myo1e/f localization in fixed cells as all cups are at slightly different stages of internalization. However, based on the Reviewer's suggestion, in ~half-way cups of transfected cells, we quantified the percentage of cells displaying myo1/actin localization as shown in Figure 3f/g: myo1e – 76% (20/26 cells) and myo1f – 71% (10/14 cells). In the few cells that didn't show this particular punctate localization within the cup, myo1 was strongly localized to the cup tip (as shown in Figure 1c/ Figure S1c). We are unsure if this difference is due to stage of internalization, cell variability, or differences in myo1 expression level (given our use of transient transfection). We are confident however that the images displayed in Figure 3f/g are not outliers.

5) On page 16, the authors say that Myo1e/f are negative regulators of the arp2/3 complex. This type of claim really requires biochemical evidence for support, and I doubt that Myo1e/f are directly opposing arp2/3 activation. It may be more precise to say that Myo1e/f corral or constrain arp2/3-mediated actin assembly.

We agree with Reviewer 3 and have changed our wording accordingly.

6) The experiments presented in figures 4 and 5 seem incomplete because they look almost exclusively at actin. Are there FcR cluster aggregates that underlie the actin aggregates?

Based on the colocalization of the FcR clusters with the actin wave puncta in RAW macrophages (Fig. 3e), we speculate that FcR clusters are indeed under the actin aggregates in the primary KO cells (as shown in our working model, Fig. 7i). We have attempted to do IHC on the primary cells with 3 different rat antibodies against FcγRII/III (CD16/32): Biolegend anti-mouse CD16/32 (clone 92), BD anti-mouse CD16/32 (clone 2.4G2), R&D FcγRI (clone 290322). Because the coverslip for the frustrated phagocytosis assay is coated in mouse antibody, the amount of background staining when detecting the rat antibody is unfortunately high. For this reason, we do not feel confident in claiming the exact localization of FcR within the primary cells during frustrated phagocytosis.

7) The SIM data in the same figure are also difficult to interpret. The wildtype cell shown doesn't seem like a good representative image, because it doesn't look like it

has a true actin wave as the wildtype cells in panel A do. Maybe the clusters in this cell would look more like a wave if the cell was imaged with conventional microscopy? If that is the case, a picture of the same cell imaged with conventional, diffraction limited microscopy should be provided.

We have replaced the SIM images of the WT and dKO cells and added an additional image. These are better, more representative images of the actin wave (Figure 4c). We do have a comparative Widefield image of the WT actin wave, prior to SIM processing, but we feel this data doesn't add much.

8) Statistical information is missing for figure 6h. How many cells were measured to generate this plot, and how many measurements were taken per cell? I'd prefer to see a more conventional scatter plot so that each data point is actually visible.

Given the response from Reviewers, previous Figure 6h has been removed and we are now simply reporting the graphed averages from Figure 6h (this graph is now Fig 6h in the current manuscript). This graph depicts data from 1 dKO macrophage undergoing dynamic actin wave retractions and protrusions. To obtain these data points, the exterior of the actin wave was first traced by hand throughout the time-lapse. From these points, wave boundary speed was determined radially by computing the distance change between two consecutive frames (actin wave protrusion was defined as a positive number, while retraction was negative). The corresponding mean actin intensity for the same angles is averaged from the wave edge boundary to 30 pixels (~4um) inward. Each data point is the measurement from each angle (0-359 degrees) throughout the time-lapse. We are including the video of the cell analyzed to assist in our explanation (see Supplementary Movie 14). Therefore, 39,131 measurements from this cell are reported, with the standard deviation error bars arising from individual measurements. We conducted the same analysis on multiple dKO cells to generate a similar trend (reported in Supplementary Figure 10). However, because cells exhibit such variability in dynamics when performing frustrated phagocytosis, using a large number of cells doesn't necessarily help us to answer the question of whether increased actin intensity in the wave correlates with reduced actin wave speed. For this we needed to analyze a dKO macrophage showing both protrusive and retractive wave behavior, however a good number of dKO macrophages only exhibit stable, almost stuck actin waves.

9) In figure 7, how big are the areas of elevated membrane that surround the actin spots? Are the actin spots always right in the middle of the elevated membrane? I would like to see more quantification of these data. At the minimum, I'd like to see a version of figure 7f where the same measurement was done on multiple cells and the results normalized and averaged. This would strengthen the case that the membrane-lifting phenomenon is actually absent in the dKO cells.

We have added more descriptive and quantitative information on the actin adhesions in the new Figure 7 (Fig. 7e-i, Supplementary Movies 15-16) and accompanying Results section (pg. 20). Figure 7f in the previous manuscript was in fact an average line scan of >80 adhesions from multiple cells to help verify our initial observations. However, we've chosen to replace this with an averaged image of membrane signal

intensity in the entire area surrounding each adhesion, for a more comprehensive analysis (Figure 7h).

Non-experimental suggestions

1) The title of this work could be more descriptive. The manuscript is mostly concerned with organization of F-actin at FcR adhesion sites, but that isn't mentioned in the title. I don't know what "membrane-cytoskeleton mechanical feedback" means. Live imaging of phagocytosis in the dKO cells (see above) may allow the authors to say something more articulate than simply saying myosin 1 "controls phagocytic efficiency."

Thank you for your suggestion. We have indeed changed the title to make it more representative of our findings.

2) A minor point: it seems strange to me that Nambiar, McConnell, and Tyska (PNAS, 2009) is not cited here, because it seems very relevant to the work.

We agree with Reviewer 3, and we were a bit surprised to find that we have indeed omitted this work. It may have gotten cut from the manuscript due to the limit on the number of references, but we have included it in the resubmission.

3) Figure 1i is not cited in the main text and it is hard to see phagocytosed beads since the image is so small. The non-phagocytosed beads should be highlighted as well. For visualization purpose it might be better to move it to the supplementary Figures and enlarge the image.

Figure 1i is cited in the main text, but by "Figure (h-i)". We would like to keep this image in the main Figure, so we have shifted things slightly and enlarged it. There appears to be no non-phagocytosed beads in this image. Most of these beads are removed prior to fixation by washing, as described in our Methods sections on the phagocytosis assay.

4) In figure 1 and supplementary figure 1 it was confusing to see part of the results of Myo1e and Myo1f mixed in both figures for the reader. I would suggest only showing the results of Myo1e or Myo1f in the main Figure and move one of the results of Myo1e or Myo1f in the supplement.

Thank you for your suggestion, we have made Figure 1 all myo1e, while myo1f is found in the Supplementary Figure.

5) On page 7 there is an error in referencing figures. The text refers to "Figure 1h-1", there is no figure 1l.

This typo has been corrected.

6) It would be helpful to summarize the findings in a model figure at the end, as was partially done in Figure 3h. That diagram is very helpful!

We hope our graphical working model (Fig. 7h) is helpful.

Reviewer #4 (Remarks to the Author):

In this work by Barger et al. The role of the myosin motors myosin 1e and myosin 1f are investigated in phagocytosis of antibody-coated particles in macrophages. The authors find that these motors which link the actin cytoskeleton to the plasma membrane are concentrated near F-actin sites that form around the phagocytic cups and FcR-actin adhesions. Double knock-outs for myosin1e/f showed slower and less uptake of 6 micron Ig-coated beads. These cells had lower membrane tensions and malformed large actin adhesions. Thus, the authors propose that the inability to form the correct actin networks in these myosin mutants could be decreasing the proper assembly of actin near FcR-based adhesions necessary for the completion of phagocytosis. In general this paper is well done, comprehensive, and interesting. The figures are quite nice and the imaging/experiments are impressive. I have several questions and comments regarding this work that might be addressed to improve the manuscript.

We thank the Reviewer for this positive evaluation.

1. In figure 1h the major (most dramatic) effect of the dKO appears at the 15 min time point and less so at the 60 min time point with only a ~30% reduction in cells with particles compared to wild-type. This effect isn't dramatic. If myosin1e/f is central for phagocytosis why the relatively minor reduction in phagocytic activity? Are there other situations where the myosin would be more critical? Clearly, the protein is involved in fine-tuning and enhancing phagocytic uptake but could the authors comment on the relative importance of these motors in the overall process?

We agree with the conclusions of Reviewer 4, in that myo1e/f are not *required* for phagocytosis, in that their removal only results in a lag in phagocytic uptake. Comparing the importance of myo1e/f to other proteins involved in phagocytosis is thus difficult when other studies use only one time point to report a reduction in phagocytosis upon protein knockout/inhibition. However, such a time-dependent defect in phagocytosis has been observed in the absence/inhibition of other proteins: myosin IIA (Olazabal et al. (2002) *Curr Bio*), WASP (Lorenzi et al. (2000) *Blood*), and Arp2/3 (Rotty et al. (2017) *Dev Cell*). We attempted to investigate the importance of myo1e/f in the overall process by dissecting and performing experiments on each step of phagocytosis. We therefore concluded the myo1e/f is critical for phagocytic adhesion turnover required for phagocytic cup closure and propose that myosin 1s may be most critical upon internalization of large targets. We have altered the Discussion to emphasize this point.

2. How is this myosin targeted to the FcR clusters? The authors note that a TH2 domain mutation is key for localization. Does this domain interact with any specific

proteins? Also, does re-expression of this mutant fail to rescue the dKO phenotype in Figure 1h? Does expression of just the TH2 domain block phagocytosis?

We have found that the TH2 domain alone localizes to the actin wave (see Supplementary Figure 6d). However, at this point we do not have specific binding partners for this domain. Since this region is predicted to be intrinsically disordered, identifying binding partners may be non-trivial. With regards to the potential of the TH2 domain to block phagocytosis, we have conducted some past preliminary experiments using RAW macrophages related to this point. Specifically, we found that RAW macrophages expressing only EGFP are significantly more phagocytic than cells expressing EGFP-m1e tail construct. However, because we are unsure of the ability of the isolated TH2 domain vs myosin 1 tail to act as dominant negative constructs in this system, we would rather not report this data. Given the time-dependent reduction in phagocytosis observed in the dKO cells (which the Reviewer noted in 1.), we speculate that expression of the TH2 domain would not completely block phagocytosis.

3. The “lifting” observation in Figure 7 is not clearly describe or explained in the text or figures. Furthermore, I cannot see how this “lifted” TIRF membrane spatially relates to the other fluorescent signals (myo or actin) shown in figure 7. Additional quantitative plotting or analysis above a single line scan that is currently shown would help this figure. As this concept is a major feature of the author’s model, and as it stands now is merely observational, it would be nice to quantitatively analyze this effect more extensively.

We have added more descriptive information and quantifications on our membrane lifting observations in Figure 7 (Fig 7e-i, Supplementary Movie 15-16), which is further described in the accompanying Results section (pg. 20). To clarify, the line scan shown in Figure 7f of the previous manuscript was in fact an average line scan from >80 adhesions in multiple cells to help verify quantitatively our initial observations. We have chosen to replace this figure with an averaged square image surrounding the adhesions (Figure 7h), again from many adhesions in multiple cells. We have also added a cartoon schematic of our working model as Figure 7i.

As this was observed in the Orci 1983 paper it would also be nice to incorporate those older models more clearly and prominently in this paper.

The conclusions from the Orci and Kaplan papers were that the electron-dense regions of focal contact between the phagocyte and the target were the morphological counterpart of the “zipper” mechanism. We mention this model briefly in the introduction, but we have now altered the Discussion to more fully include it (pg. 22). Thank you for the suggestion.

4. The paper would benefit from a general cartoon figure where the authors clearly propose their model of adhesion and cup formation with the “lifting” idea and the role of these myosins in that process.

Thank you, we have included such a Figure (7i).

5. Why is the initial phagocytic cup not impacted by myosin dKO but the later stages (closure?) of phagocytosis impacted? If the adhesion/tension and lifting behavior is key I would expect a more pronounced change throughout phagocytosis. Some discussion or model for this part of the process would benefit the paper and help with future experimental tests of this model.

This is a good question and one we hope to have answered more clearly in the writing of our resubmitted manuscript. Based on our experimental model of 2D frustrated phagocytosis, we observe that initial cell spreading (i.e. when a macrophage first encounters a coated substrate) is fast with only sparse punctate adhesions. The rate of this spreading does not differ between WT and dKO macrophages (Supplementary Figure 4h). We estimate that this fast membrane extension period roughly corresponds to the extension of the phagocytic cup formed halfway. In accordance with this speculation, at an earlier time point, WT and dKO macrophages both seem to form phagocytic cups at a similar rate (Figure 1k). However, we think that when it comes to closing the phagocytic cup, especially in the case of large targets, phagocytic adhesion turnover is critical. This is when the timely adhesion dynamics of WT cells (see Figure 6a) seems to overcome the sluggish dynamics of the dKO cells to complete internalization. This is our working model as illustrated in Figure 7i.

Reviewers' Comments:

Reviewer #2:

Remarks to the Author:

All of my concerns have been addressed and the manuscript has been significantly improved. The manuscript is a significant contribution and worthy of publication in Nature Communications.

Reviewer #3:

Remarks to the Author:

The authors have adequately addressed my concerns.

Reviewer #4:

Remarks to the Author:

The authors have addressed all my specific concerns and the new revised manuscript is much improved and quite impressive. I think it will be a useful addition to this field from both a molecular and structural cell biology perspective.